# Asymptotically Stable Quaternion-valued Hopfield-structured Neural Networks with Periodic Projection-based Supervised Learning Rules

**Tianwei Wang**[†]
University of Edinburgh
T.Wang-110@sms.ed.ac.uk

**Xinhui Ma**
University of Hull
xinhui.ma@hull.ac.uk

**Wei Pang**[†§]
Heriot-Watt University
w.pang@hw.ac.uk

## Abstract

Motivated by the geometric advantages of quaternions in representing rotations and postures, we propose a quaternion-valued supervised learning Hopfield-structured neural network (QSHNN) with a fully connected structure inspired by the classic Hopfield neural network (HNN). Starting from a continuous-time dynamical model of HNNs, we extend the formulation to the quaternionic domain and establish the existence and uniqueness of fixed points with asymptotic stability. For the learning rules, we introduce a periodic projection strategy that modifies standard gradient descent by periodically projecting each $4 \times 4$ block of the weight matrix onto the closest quaternionic structure in the least-squares sense. This approach preserves both convergence and quaternionic consistency throughout training. Benefiting from this rigorous mathematical foundation, the experimental model implementation achieves high accuracy, fast convergence, and strong reliability across randomly generated target sets. Moreover, the evolution trajectories of the QSHNN exhibit well-bounded curvature, i.e., sufficient smoothness, which is crucial for applications such as control systems or path planning modules in robotic arms, where joint postures are parameterized by quaternion neurons. Beyond these application scenarios, the proposed model offers a practical implementation framework and a general mathematical methodology for designing neural networks under hypercomplex or non-commutative algebraic structures.

## 1 Introduction

Hopfield neural network (HNN) could be recognized as the precursor of many fundamental models [27, 33, 10]. It has a symmetric network topology and recurrent connections, becoming an attractor model with a dynamical structure, where neurons are activated and interact with each other at a certain frequency and tend to discrete equilibria. These equilibria and the attracting landscape in phase space embody the generalization capability of the neural network. We adopt the continuous time HNN [11] governed by the evolution Eq. (1.1) to expand and rebuild QSHNN in this research.

$$\frac{\mathrm{d}v_j(t)}{\mathrm{d}t} = -\gamma v_j(t) + \mu \sum_{i=1}^{n} w_{ji}\varphi\{v_i(t)\} + \mu b_j, \quad j = 1, 2, ...n \quad (1.1)$$

In a more condensed vector form $\dot{\boldsymbol{v}}(t) = -\gamma \boldsymbol{v}(t) + \mu W \boldsymbol{\varphi}(\boldsymbol{v}) + \mu \boldsymbol{b}$, Vector $\boldsymbol{v}$ represents the list of the neuron activations $\boldsymbol{v}_j$, matrix $W$ arranges the connection weights $\boldsymbol{w}_{ji}$ in matrix-vector multiplication, vector function $\boldsymbol{\varphi}$ imposes the activating function, typically hyperbolic tangent $\varphi(x) = (e^x - e^{-x})/(e^x + e^{-x})$, to each entry of vector $\boldsymbol{v}$. $\boldsymbol{b}$ serves as the bias vector and is significant to the representation ability of the network. Meaning of constant $\gamma, \mu$ in corresponds

---

[†]Bio-inspired Computing and Machine Learning (BCML) Lab.
[§]Corresponding author

39th Conference on Neural Information Processing Systems (NeurIPS 2025).

to the parameters of electronic components capacitance and resistor for the circuit implementation of classic HNN. The stability of HNNs is fundamental to their function as associative memory systems, enabling the convergence of network states to attractor patterns [11]. Earlier studies have extended HNNs to quaternion-valued systems, mainly under discrete-time formulations with split activation functions and fixed-point dynamics [9, 15]. More recently, continuous-time modern HNNs have also been introduced [23, 1], where the state evolution follows ordinary differential equations. However, these models remain largely confined to unsupervised paradigms driven by internal energy minimization, lacking explicit target tracking, structural control, or generalized learning principles.

In contrast, our proposed model departs from this associative memory paradigm of HNN. We introduce a quaternion-valued supervised Hopfield-structured neural network (QSHNN), formulated as a continuous-time autonomous dynamical system. The system's state evolves according to a quaternionic differential equation, and the network is trained to converge asymptotically to externally specified targets. The learning rule is derived analytically via Generalized $\mathbb{HR}$ (GHR) calculus [36] and incorporates a projection mechanism to preserve the block-wise quaternion structure of the weight matrix. GHR calculus has been established and verified as a complete and mature technique to expand the neural network over the quaternionic domain [21, 38, 37, 5, 22, 29], where the normal differential formulas are no longer valid because of the non-commutativity of quaternion algebra. This design guarantees both convergence and structural consistency.

Although early Hopfield-type networks, including their quaternion-valued extensions, often rely on directly encoding the weight matrix using Hebbian or outer-product formulations, this approach suffers from several critical limitations. First, such weight constructions inherently lack scalability; they can only stably store a small number of target states before spurious attractors emerge or convergence fails [28, 4, 19]. Second, these methods assume all target patterns are fixed and known a *priori*, making the network unsuitable for tasks requiring adaptability, generalization, or dynamic reconfiguration [3, 19]. Third, direct encoding lacks an error-driven optimization mechanism, preventing the network from refining its behavior in response to task-specific objectives [19, 17]. As a result, while analytically convenient, direct weight specification severely restricts the practical applicability of Hopfield-type models in real-world control, learning, or representation settings. Unlike classical or previously proposed quaternion-valued Hopfield neural networks [12, 14, 32], our model operates under a supervised learning paradigm with continuous quaternion-valued trajectories, offering smooth, controllable dynamics suitable for robotics, trajectory generation, and feedback control systems. We position this work not as a generalization of existing QHNNs, but as the formulation of a new class of quaternionic neural dynamical systems with rigorous theoretical guarantees and task-driven behavior.

## 2 Background

**Quaternion algebra**  Quaternion is a hypercomplex number with three imaginary components; we separate the components into the scalar part $s$ and the vector part $\boldsymbol{v}$: $\boldsymbol{q} = s + x\boldsymbol{i} + y\boldsymbol{j} + z\boldsymbol{k} = [s, \boldsymbol{v}] \in \mathbb{H}$. The vector part space matches the 3-dimensional Euclidean space $\mathbb{R}^3$. The region of quaternions is denoted by $\mathbb{H}$, where the coefficients associated with the imaginary units are $x, y, z \in \mathbb{R}$. The fundamental arithmetic of imaginary units are $\boldsymbol{i} \circ \boldsymbol{j} = -\boldsymbol{j} \circ \boldsymbol{i} = \boldsymbol{k}, \quad \boldsymbol{j} \circ \boldsymbol{k} = -\boldsymbol{k} \circ \boldsymbol{j} = \boldsymbol{i}, \quad \boldsymbol{k} \circ \boldsymbol{i} = -\boldsymbol{i} \circ \boldsymbol{k} = \boldsymbol{j}, \boldsymbol{i}^2 = \boldsymbol{j}^2 = \boldsymbol{k}^2 = \boldsymbol{i} \circ \boldsymbol{j} \circ \boldsymbol{k} = -1$.

We denote the multiplication operation of two quaternions $\boldsymbol{q}_1$ and $\boldsymbol{q}_2$ by $\boldsymbol{q}_1 \circ \boldsymbol{q}_2$. To distinguish other types of multiplication, we neglect the symbol for matrix multiplication and scalar multiplication. We denote the inner product by $\boldsymbol{v}_1 \cdot \boldsymbol{v}_2$, and the cross product by $\boldsymbol{v}_1 \times \boldsymbol{v}_2$. Consider quaternion $\boldsymbol{q_1} = s_1 + x_1\boldsymbol{i} + y_1\boldsymbol{j} + z_1\boldsymbol{k}$ and $\boldsymbol{q} = s_2 + x_2\boldsymbol{i} + y_2\boldsymbol{j} + z_2\boldsymbol{k}$, the plain expansion of multiplication becomes: $\boldsymbol{q_1} \circ \boldsymbol{q_2} = (s_1 s_2 - x_1 x_2 - y_1 y_2 - z_1 z_2) + (s_1 x_2 + s_2 x_1 + y_1 z_2 - y_2 z_1)\boldsymbol{i} + (s_1 y_2 + s_2 y_1 + z_1 x_2 - z_2 x_1)\boldsymbol{j} + (s_1 z_2 + s_2 z_1 + x_1 y_2 - x_2 y_1)\boldsymbol{k}$.

**Quaternion left multiplication manifold**  Quaternion multiplication can be represented as real-valued matrix–vector multiplication over $\mathbb{R}^4$ by Eq. 2.1), where each quaternion defines a $4 \times 4$ real matrix that acts on another quaternion interpreted as a 4-dimensional real vector. Due to the non-commutative nature of quaternion algebra, there exist two distinct but equivalent matrix representations: one for left multiplication and one for right multiplication. In this work, we adopt the left multiplication form consistently, where each quaternion induces a left-linear action on the

quaternionic space. This operation is denoted by $\circ$ between two quaternions with the following form:

$$\boldsymbol{q_1} \circ \boldsymbol{q_2} = (s_1 + x_1\boldsymbol{i} + y_1\boldsymbol{j} + z_1\boldsymbol{k}) \circ (s_2 + x_2\boldsymbol{i} + y_2\boldsymbol{j} + z_2\boldsymbol{k})$$

$$= \begin{bmatrix} s_1 & -x_1 & -y_1 & -z_1 \\ x_1 & s_1 & -z_1 & y_1 \\ y_1 & z_1 & s_1 & -x_1 \\ z_1 & -y_1 & x_1 & s_1 \end{bmatrix} \begin{bmatrix} s_2 \\ x_2 \\ y_2 \\ z_2 \end{bmatrix} = \begin{bmatrix} s_2 & -x_2 & -y_2 & -z_2 \\ x_2 & s_2 & z_2 & -y_2 \\ y_2 & -z_2 & s_2 & x_2 \\ z_2 & y_2 & -x_2 & s_2 \end{bmatrix} \begin{bmatrix} s_1 \\ x_1 \\ y_1 \\ z_1 \end{bmatrix} \qquad (2.1)$$

The collection of all real $4 \times 4$ matrices corresponding to quaternion left multiplication forms a smooth 4-dimensional embedded submanifold of $\mathbb{R}^{4\times4}$, denoted by $\mathcal{L}$. This submanifold inherits a natural algebraic structure from $\mathbb{H}$ and constitutes a real matrix Lie group under composition [30, 25]. Specifically, $\mathcal{L}$ is isomorphic to the group $(\mathbb{H}, \circ)$, and the group operation corresponds to matrix multiplication within $\mathcal{L}$. Each matrix $L(\boldsymbol{q}) \in \mathcal{L}$ is uniquely determined by a quaternion $\boldsymbol{q} = s + x\boldsymbol{i} + y\boldsymbol{j} + z\boldsymbol{k}$, and the map $\boldsymbol{q} \mapsto L(\boldsymbol{q})$ is a group isomorphism. The tangent space at the identity element L(1) defines the associated Lie algebra, which consists of all real $4 \times 4$ matrices that can be written as linear combinations of the infinitesimal generators $L(\boldsymbol{i}), L(\boldsymbol{j}), L(\boldsymbol{k})$.

$$L(1) = \mathbb{I}_4, \; L(i) = \begin{bmatrix} 0 & 1 & 0 & 0 \\ - & 0 & 0 & 0 \\ 0 & 0 & 0 & 1 \\ 0 & 0 & - & 0 \end{bmatrix}, \; L(j) = \begin{bmatrix} 0 & 0 & 1 & 0 \\ 0 & 0 & 0 & - \\ - & 0 & 0 & 0 \\ 0 & 1 & 0 & 0 \end{bmatrix}, \; L(k) = \begin{bmatrix} 0 & 0 & 0 & 1 \\ 0 & 0 & 1 & 0 \\ 0 & - & 0 & 0 \\ - & 0 & 0 & 0 \end{bmatrix}. \qquad (2.2)$$

Each element of the quaternion left multiplication manifold $\mathcal{L}$ can be expressed as a linear combination of four fixed basis matrices corresponding to the canonical quaternion basis $\{1, \boldsymbol{i}, \boldsymbol{j}, \boldsymbol{k}\}$. Specifically, for any quaternion $q = s + x\boldsymbol{i} + y\boldsymbol{j} + z\boldsymbol{k} \in \mathbb{H}$, its corresponding left multiplication matrix $L(q) \in \mathcal{L}$ admits the decomposition: $L(q) = s\,L(1) + x\,L(\boldsymbol{i}) + y\,L(\boldsymbol{j}) + z\,L(\boldsymbol{k})$, where $L(1), L(\boldsymbol{i}), L(\boldsymbol{j}), L(\boldsymbol{k})$ are fixed real $4 \times 4$ matrices, as shown in Eq. (2.2), representing left multiplication by each of the basis elements. This decomposition establishes a linear isomorphism between $\mathbb{H}$ and subspace $\mathcal{L}$.

And we also define the conjugate of a quaternion by $\boldsymbol{q}^\dagger = s - x\boldsymbol{i} - y\boldsymbol{j} - z\boldsymbol{k} = [s, -\boldsymbol{v}]$, the modulus or length of a quaternion by $|\boldsymbol{q}| = \sqrt{\boldsymbol{q} \circ \boldsymbol{q}^\dagger} = (s_{\boldsymbol{q}}^2 + x_{\boldsymbol{q}}^2 + y_{\boldsymbol{q}}^2 + z_{\boldsymbol{q}}^2)^{1/2}$, the inverse of a quaternion by $\boldsymbol{q} \circ \boldsymbol{q}^{-1} = 1$, where $\boldsymbol{q}^{-1} = \boldsymbol{q}^\dagger/|\boldsymbol{q}|$. It is quite important to mention the geometric meaning and advantages of quaternion, which is just one of our motivations. There is a compact and intuitive form to represent 3-dimensional rotation by a quaternion.

**Definition 2.1** (Quaternion rotation [34]). *Rotating an arbitrary quaternion $\boldsymbol{q} = s_{\boldsymbol{q}} + \boldsymbol{v}_{\boldsymbol{q}}$ by a quaternion number parameterized by $\boldsymbol{\mu} = |\mu|(cos\beta + \boldsymbol{v}_{\boldsymbol{\mu}}sin\beta)$ is equivalent to computing the following:*

$$R_{\boldsymbol{\mu}}(\boldsymbol{q}) := \boldsymbol{\mu} \circ \boldsymbol{q} \circ \boldsymbol{\mu}^{-1} \equiv \boldsymbol{q}^{\boldsymbol{\mu}} \qquad (2.3)$$

*That is to make the 3-dimensional vector $\boldsymbol{v}_{\boldsymbol{q}}$ rotate by angle $\beta$ about the axis $\boldsymbol{v}_{\boldsymbol{\mu}}$. When $\mu$ is a pure unit quaternion $(\boldsymbol{i}, \boldsymbol{j}, \boldsymbol{k})$, this manipulation is also named quaternion involution.*

As a non-commutative algebra, quaternion is not compatible with general calculus theory. Hence, an extension of differential is necessary to utilize the derivative of quaternion as well as the chain rule, product rule, etc., for the practical purpose of theoretical deduction.

**Definition 2.2** (Quaternion (left GHR) derivative [36]). *Let $\boldsymbol{q} \in \mathbb{H}$ and $\boldsymbol{f} : D \to \mathbb{H}$, $D \subseteq \mathbb{H}$, then the left GHR derivatives, with respect to $\boldsymbol{q}^{\boldsymbol{\mu}}$ and $\boldsymbol{q}^{\boldsymbol{\mu}*}$ ($\boldsymbol{\mu} \neq 0, \boldsymbol{\mu} \in \mathbb{H}$) of a well-defined function $\boldsymbol{f}$ are defined as:*

$$\frac{\partial \boldsymbol{f}}{\partial \boldsymbol{q}^{\boldsymbol{\mu}}} = \frac{1}{4}\left(\frac{\partial \boldsymbol{f}}{\partial \boldsymbol{q}_a} - \frac{\partial \boldsymbol{f}}{\partial \boldsymbol{q}_b} \circ \boldsymbol{i}^{\boldsymbol{\mu}} - \frac{\partial \boldsymbol{f}}{\partial \boldsymbol{q}_c} \circ \boldsymbol{j}^{\boldsymbol{\mu}} - \frac{\partial \boldsymbol{f}}{\partial \boldsymbol{q}_d} \circ \boldsymbol{k}^{\boldsymbol{\mu}}\right) \qquad (2.4)$$

$$\frac{\partial \boldsymbol{f}}{\partial \boldsymbol{q}^{\boldsymbol{\mu}\dagger}} = \frac{1}{4}\left(\frac{\partial \boldsymbol{f}}{\partial \boldsymbol{q}_a} + \frac{\partial \boldsymbol{f}}{\partial \boldsymbol{q}_b} \circ \boldsymbol{i}^{\boldsymbol{\mu}} + \frac{\partial \boldsymbol{f}}{\partial \boldsymbol{q}_c} \circ \boldsymbol{j}^{\boldsymbol{\mu}} + \frac{\partial \boldsymbol{f}}{\partial \boldsymbol{q}_d} \circ \boldsymbol{k}^{\boldsymbol{\mu}}\right) \qquad (2.5)$$

*where $\frac{\partial \boldsymbol{f}}{\partial \boldsymbol{q}_a}$, $\frac{\partial \boldsymbol{f}}{\partial \boldsymbol{q}_b}$, $\frac{\partial \boldsymbol{f}}{\partial \boldsymbol{q}_c}$, and $\frac{\partial \boldsymbol{f}}{\partial \boldsymbol{q}_d}$ are the partial derivatives of $\boldsymbol{f}$ with respect to $\boldsymbol{q}_a$, $\boldsymbol{q}_b$, $\boldsymbol{q}_c$, and $\boldsymbol{q}_d$, respectively, and $\{1, \boldsymbol{i}^{\boldsymbol{\mu}}, \boldsymbol{j}^{\boldsymbol{\mu}}, \boldsymbol{k}^{\boldsymbol{\mu}}\}$ is an orthogonal basis of $\mathbb{H}$ as it has the same rotation factor $\boldsymbol{\mu}$.*

**Lyapunov stability theory** For the unknown solution to the evolution equation, which is of high-level nonlinearity and may not even exist in analytical form, we adopt Lyapunov theory for the analysis of stability. There are three different types of stability for a dynamical system that are in our consideration: Lyapunov stability, quasi-asymptotic stability, and asymptotic stability. The justification can be drawn from directly using the following Thm. (2.1).

**Theorem 2.1** (Lyapunov Theorem [26])**.** *Consider the dynamical system of the form $\dot{\boldsymbol{x}} = \boldsymbol{f}(\boldsymbol{x}, t)$ with the fixed point at the origin, where $\boldsymbol{f}(\boldsymbol{0}, t) \equiv \boldsymbol{0}$. If there exists a positive definite scalar function $V(\boldsymbol{x}, t)$, also named the Lyapunov energy function, such that the time derivative $\frac{dV}{dt}$ on the flow is semi-negative definite, as shown in Eq. (2.6), we have the zero equilibrium being Lyapunov stable.*

$$\frac{dV(\boldsymbol{x}, t)}{dt} = \frac{\partial V}{\partial t} + \frac{\partial V}{\partial \boldsymbol{x}} \boldsymbol{f}(\boldsymbol{x}, t) \leq 0 \tag{2.6}$$

*Further, if the time derivative on the flow is strictly negative definite, then we have that the fixed point is asymptotically stable. The critical step in applying this theorem is to find a suitable energy function.*

## 3  Methodology

**Structure of quaternion neuron**   To allow quaternion numbers to operate on the neural network, we integrate four real-valued neurons as a quaternion neuron and impose specific internal evolution regulations, as shown in Fig. (1). A single neuron contains four normal Hopfield neurons, each of them represents a division component of a quaternion. Input vector will be $\boldsymbol{q}_{input}$ and output vector will be $\boldsymbol{q}_{out}$. With a unit time delay, the output value loops into the neuron and is weighted, then activates the linked neuron by a function $\varphi$. Every connection is allocated a weight $w_{ij}$, and four weights are integrated as a quaternion weight $\boldsymbol{\omega}$.

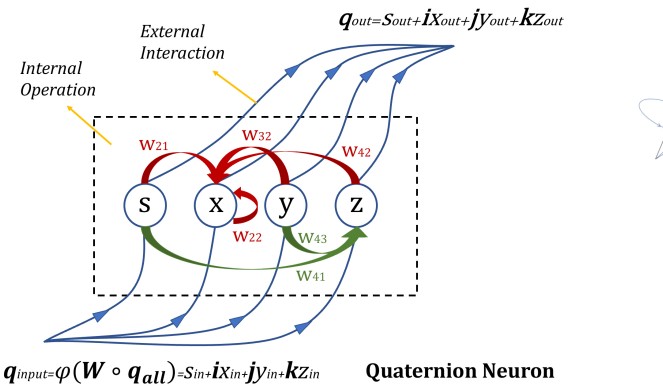

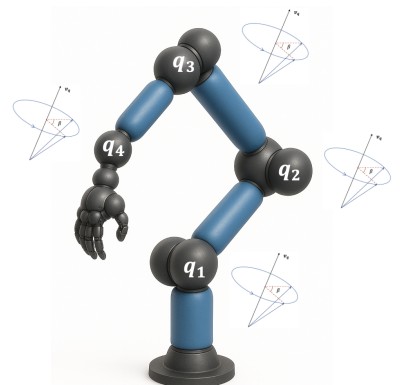

Figure 1: Structure of quaternion neuron and its interfaces. Four real-valued neurons are integrated together as a quaternion with respect to different component coefficients. The internal operation or the self-connection can be fully characterized by a single weight quaternion. Every numerical factors are compatible by quaternions, projection and approximation only exist in the training. Drawn by the authors.

Figure 2: Robotic arm with each joint posture parameterized by quaternions for each joint. Our model is capable of high fidelity control and path planning. Figure drawn by the authors, where the robotic arm is generated by AI tool just for demonstration aim.

**Consistency and compatibility of mathematical form**   Quaternion neuron structure is closely related to the mathematical form of the evolution equation, where we embed this structure into the classic HNN evolution Eq. (1.1). The quaternion-valued system is consistent with the original form by directly replacing the real variables with quaternionic variables through Eq. (2.1). Explicitly, it can be expressed in quaternion algebra and underlying linear algebra by:

$$\frac{d}{dt}\begin{bmatrix}\boldsymbol{q}_1\\\boldsymbol{q}_2\\\vdots\\\boldsymbol{q}_n\end{bmatrix} = -\gamma\begin{bmatrix}\boldsymbol{q}_1\\\boldsymbol{q}_2\\\vdots\\\boldsymbol{q}_n\end{bmatrix} + \mu\begin{bmatrix}\boldsymbol{\omega}_{11}&\boldsymbol{\omega}_{12}&\cdots&\boldsymbol{\omega}_{1n}\\\boldsymbol{\omega}_{21}&\boldsymbol{\omega}_{22}&\cdots&\boldsymbol{\omega}_{2n}\\\vdots&\vdots&\ddots&\vdots\\\boldsymbol{\omega}_{n1}&\boldsymbol{\omega}_{n2}&\cdots&\boldsymbol{\omega}_{nn}\end{bmatrix} \circ \begin{bmatrix}\varphi(\boldsymbol{q_1})\\\varphi(\boldsymbol{q_2})\\\vdots\\\varphi(\boldsymbol{q_n})\end{bmatrix} + \mu\begin{bmatrix}\boldsymbol{b}_1\\\boldsymbol{b}_2\\\vdots\\\boldsymbol{b}_n\end{bmatrix} \tag{3.1}$$

$$= -\gamma\begin{bmatrix}q_1\\q_2\\\vdots\\q_{4n}\end{bmatrix} + \mu\begin{bmatrix}W_{11}&W_{12}&\cdots&W_{1n}\\W_{21}&W_{22}&\cdots&W_{2n}\\\vdots&\vdots&\ddots&\vdots\\W_{n1}&W_{n2}&\cdots&W_{nn}\end{bmatrix}\begin{bmatrix}\varphi(q_1)\\\varphi(q_2)\\\vdots\\\varphi(q_{4n})\end{bmatrix} + \mu\begin{bmatrix}b_1\\b_2\\\vdots\\b_{4n}\end{bmatrix} \tag{3.2}$$

Or in a condensed form $\dot{\boldsymbol{q}} = -\gamma \boldsymbol{q} + \mu \boldsymbol{W} \circ \boldsymbol{\varphi}(\boldsymbol{q}) + \mu \boldsymbol{b} \equiv -\gamma \boldsymbol{q} + \mu W \varphi(\boldsymbol{q}) + \mu \boldsymbol{b}$, where we denote the quaternionic weight from neuron $i$ to neuron $j$ by $\boldsymbol{\omega}_{ji}$, and arrange all the connections as a quaternion matrix $\boldsymbol{W}$. Quaternionic matrix-vector multiplication is denoted by $\boldsymbol{W} \circ \boldsymbol{q}$.

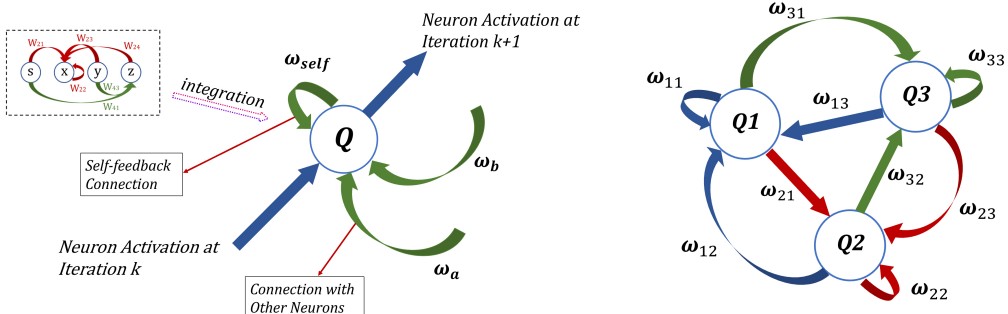

Figure 3: The integration of quaternion neuron. It has inside structure, self feedback connection and connection interfaces with other neurons. Activation will be updated every unit impulse time. Drawn by the authors.

Figure 4: Structure of a fully connected 3-neuron network. Every connection from j-th to i-th neuron is parameterized directly by a quaternion $\omega_{ij}$. Drawn by authors.

**Single neuron response with linear activation**   In this subsection, we state the linear response properties of the neuron structure determined above. Then we discuss the activation function we choose for our model eventually. For a single neural feedback with linear activation $\varphi : \boldsymbol{q} \to \boldsymbol{q}$, the evolution equation becomes:

$$\dot{\boldsymbol{q}}(t) = -\gamma \boldsymbol{q} + \mu \boldsymbol{\omega} \circ \boldsymbol{q} + \mu \boldsymbol{b} := \boldsymbol{\chi} \circ \boldsymbol{q} + \mu \boldsymbol{b} \equiv X \boldsymbol{q} + \mu \boldsymbol{b} \tag{3.3}$$

where the new quaternion coefficient $\boldsymbol{\chi} = -\gamma + \mu \boldsymbol{\omega}$ with respect to the left multiplication matrix $X$. Thus, it becomes a linear differential system with a nonhomogeneous term $\mu \boldsymbol{b}$. The solution of the homogeneous equation can be calculated directly. Through the matrix method, firstly, we define the fundamental matrix by $\Phi(t) = [\boldsymbol{\zeta_1} \ \boldsymbol{\zeta_2} \ \boldsymbol{\zeta_3} \ \boldsymbol{\zeta_4}]$. $X$ is noticeable as an anti-symmetric matrix plus a scalar times the identity matrix $E$.

Notice that this anti-symmetric matrix has only imaginary eigenvalues, which makes it easy to know $X$ has two different conjugate complex eigenvalues, and they all have the same algebraic multiplicity, and the geometric multiplicity $m_g = 1$, corresponding to complex eigenvalues $\lambda = \pm \chi_0 \mp \boldsymbol{i}(\chi_x^2 + \chi_y^2 + \chi_z^2)^{\frac{1}{2}}$. To make the formula compact, the square root of the square sum in terms of weight parameters is denoted by $\alpha^2 = \chi_x^2 + \chi_y^2 + \chi_z^2 = \beta^{-2}$. Then we calculate the exponential matrix, and it still has the form of a quaternionic anti-symmetric matrix. Through the adjacent matrix, the matrix exponential associated with $\Phi$ is calculated by:

$$e^{Xt} = \Phi(t)\Phi(0)^{-1} = \frac{e^{\chi_0 t}}{\alpha} \cdot \begin{bmatrix} \beta \cos \alpha t & -\chi_x \sin \alpha t \alpha & -\chi_y \sin \alpha t & -\chi_z \sin \alpha t \\ \chi_x \sin \alpha t & \beta \cos \alpha t & -\chi_z \sin \alpha t & \chi_y \sin \alpha t \\ \chi_y \sin \alpha t & \chi_z \sin \alpha t & \beta \cos \alpha t & -\chi_x \sin \alpha t \\ \chi_z \sin \alpha t & -\chi_y \sin \alpha t & \chi_x \sin \alpha t & \beta \cos \alpha t \end{bmatrix} \tag{3.4}$$

$$= e^{\chi_0 t} \cos \alpha t + \frac{\chi_x \sin \alpha t}{\alpha} \boldsymbol{i} + \frac{\chi_y \sin \alpha t}{\alpha} \boldsymbol{j} + \frac{\chi_z \sin \alpha t}{\alpha} \boldsymbol{k} := \boldsymbol{q}_e.$$

With the exponential of coefficient matrix and the initial activation $\boldsymbol{q}_0 = s_0 + x_0 \boldsymbol{i} + y_0 \boldsymbol{j} + z_0 \boldsymbol{k}$, we can write the general solution to the single neuron response to:

$$\boldsymbol{q}(t) = e^{Xt} \boldsymbol{q}_0 + \mu e^{Xt} \left[ \int_0^t (e^{X\tau})^{-1} d\tau \right] \boldsymbol{b} = e^{Xt} \boldsymbol{q}_0 + \mu X^{-1}(e^{Xt} - \mathbb{I}_4)\boldsymbol{b} \tag{3.5}$$

$$\equiv \boldsymbol{q}_e \circ \boldsymbol{q}_0 + \mu \boldsymbol{\chi}^\dagger \circ (\boldsymbol{q}_e - 1) \circ \boldsymbol{b}$$

The entire expression perfectly integrates quaternion embeddings instead of through complex mapping and splitting methods, where many required formalizations can be satisfied automatically. For higher order neuron systems with nonlinear activation functions, the solution would be too complicated and nonanalytic, shown in [18, 16], and machine learning approaches get involved to make a qualitative control of this neural differential equation system.

**Asymptotic Stability Analysis**    For the type of neural networks as a nonlinear dynamical system, the evolution equation is demanded to be equipped with a series of good properties. The two most fundamental of them are convergence and stability, which guarantee a deterministic activating value of the neuron and that the neuron can correctly evolve to that value, respectively. Below we propose two significant theorems guaranteeing the asymptotic stability of the QSHNN model. Firstly, we demand the existence and uniqueness of the dynamical system; they are the foundation of further analysis and critical for the capability of memory, and the stored memory will not be confused.

**Theorem 3.1** (First weights constraint of QSHNN). *For the differential dynamical system 3.2, the existence and uniqueness of equilibrium are sufficient to be proven by the following inequality on weight parameters. $L_i$ is the Lipschitz constant of the activation function on the i-th variable. $\mu$ and $\gamma$ are constants consistent with former notations.*

$$\frac{\mu}{\gamma} \sum_{i=1}^{n} L_i |w_{ij}| < 1 \tag{3.6}$$

The proof can be found in App. A. If we choose the hyperbolic tangent activation function everywhere, which has a Lipschitz constant of 1, and set constant $\mu = \gamma = 1$, then the weights constraint in Thm. 3.1 is necessary by doing the normalization on the infinite norm of matrix $W$ by $||W||_\infty = 1 - \varepsilon$. Here $\epsilon$ is a small constant to avoid the equality in the inequality 3.7. For the second significant property, we derive the asymptotic stability based on the Lyapunov theorem 2.1. With the following theorem, our model can be qualified with an asymptotically stable equilibrium.

**Theorem 3.2** (Second weights constraint of QSHNN). *For the differential dynamical system 3.2, the equilibrium that is asymptotically stable can be established by the following inequality. $\mu$ and $\gamma$ are constants consistent with former notations.*

$$\frac{\mu}{2\gamma} \sum_{i=1}^{n} (|w_{ji}| + |w_{ij}|) < 1 \tag{3.7}$$

The proof can be found in App. B. Similarly, we have a natural way to satisfy this inequality, which is sufficient to be proved after the process of normalization in training by $||W||_\infty = 1 - \varepsilon$. Thm. 3.1 and Thm. 3.2 guarantee that the network dynamics evolve to the designated target consistently. The absolute error between the target and the flow also converges with rigorous deduction in App. G.

**Smoothness of trajectories**    Let $\boldsymbol{q}(t)$ denote the state trajectory generated by the proposed QSHNN under the network dynamics Eq. (3.2). Because the activation $\varphi(\cdot)$ is globally Lipschitz continuous with constant $L_\varphi = 1$, and the weights are normalized by $||W||_\infty = 1 - \varepsilon < 1$, we obtain the curvature of the trajectories: $\kappa_i(t) = |\ddot{q}_i(t)| \cdot [1 + (\dot{q}_i(t))^2]^{-\frac{3}{2}} \leq ||\ddot{\boldsymbol{q}}||_\infty \leq 4$. App. D gives the detailed derivation and curvature evaluation comes from [7]. The upper bound $\kappa \leq 4$ follows from the choice of constants $\gamma = \mu = L_\varphi = 1$. Hence, every component of $\boldsymbol{q}(t)$ possesses a uniformly bounded second derivative. In particular, the curvature $\kappa_i(t)$ remains finite for all quaternion elements $q_i$ and $t$, guaranteeing the smoothness of the planned path on each joint motion of robotic manipulation. This level of regularity is sufficient for robotic posture control since actuator commands derived from $\dot{\boldsymbol{q}}$ are free of discontinuities, and bounded curvature precludes abrupt changes in end-effector acceleration.

**Learning rules for QSHNN.**    The learning of neural networks could follow the Hebbian learning rule [10] like HNN, which is unsupervised, local, and physically inspired, different from constructing and minimizing the loss function, or the Delta learning rule [24, 35] which is supervised, global, and optimization-driven. By taking the differential of the sensitivity function, we derive the gradient direction in the weights parameter space for the loss function. This is the first stage of our model, for that the trained weights do not preserve quaternion structure, where we demand every $4 \times 4$ block of the weights matrix is in the quaternion left multiplication submanifold. Hence we name the model supervised learning Hopfield-structured neural network (SHNN). The critical properties such as network stability and training accuracy are guaranteed with a rigorous mathematical mechanism through Thm. 3.1 and 3.2. For the second stage, we apply a technical skill called periodic projection. Every five iterations of strict gradient descent, we do a Frobenius orthogonal projection, as shown in Thm. (3.3), which may violate the principle of minimizing but imposes quaternionic structure on weight blocks. Globally, it will generate a periodic fluctuating path tending to the minimizer.

**Sensitivity equation and strict gradient descent.**   Now we start the establishment of learning rules from rigorous mathematics. In steady state we have the sensitivity equation, by setting the system Eq. (3.2) to zero:

$$\gamma \boldsymbol{q}^* = \mu W \boldsymbol{\varphi}(\boldsymbol{q}^*) + \mu \boldsymbol{b}, \tag{3.8}$$

where $\boldsymbol{q}, \boldsymbol{q}_d, \boldsymbol{b} \in \mathbb{R}^{4n}$, weights matrix $W \in \mathbb{R}^{4n \times 4n}$, and the hyperbolic tangent activation function $\boldsymbol{\varphi}(\boldsymbol{q}) = [\varphi(q_1), \dots, \varphi(q_{4n})]^T$. Differentiating the steady-state equation with respect to the element $w_{ij}$ of the weights matrix $W$ with row entry $i$ and column entry $j$ (the bias $\boldsymbol{b}$ is treated as a constant vector) by the chain rule:

$$\gamma \frac{\partial \boldsymbol{q}^*}{\partial w_{ij}} = \mu \left[ W \cdot \frac{\partial \boldsymbol{\varphi}(\boldsymbol{q}^*)}{\partial \boldsymbol{q}^*} \cdot \frac{\partial \boldsymbol{q}^*}{\partial w_{ij}} + \boldsymbol{e}_i \varphi(q_j^*) \right] \tag{3.9}$$

where $\boldsymbol{e}_i$ is the i-th standard base of vector space $\mathbb{R}^{4n \times 4n}$. Yields the sensitivity relation:

$$\frac{\partial \boldsymbol{q}^*}{\partial w_{ij}} = \left[ \mathbb{I}_{4n} - \frac{\mu}{\gamma} W \cdot J_{\boldsymbol{\varphi}}(\boldsymbol{q}^*) \right]^{-1} \frac{\mu}{\gamma} \varphi(q_j^*) \boldsymbol{e}_i, \tag{3.10}$$

where $J_{\boldsymbol{\varphi}}(\boldsymbol{q}) = \mathrm{diag}\left[\varphi'(q_1), \dots, \varphi'(q_n)\right]$, and $\mathbb{I}$ is a identity matrix of a compatible size. For the quadratic loss function defined as $E(\boldsymbol{q}) = \frac{1}{2}\|\boldsymbol{q} - \boldsymbol{q}_d\|_2$. Aiming to derive the gradient direction of the loss function about the weights. Substituting this sensitivity into the chain-rule expression gives the complete gradient formula:

$$\frac{\partial E}{\partial w_{ij}} = \frac{\partial E}{\partial (\boldsymbol{q}^* - \boldsymbol{q}_d)} \cdot \frac{\partial (\boldsymbol{q}^* - \boldsymbol{q}_d)}{\partial \boldsymbol{q}^*} \cdot \frac{\partial \boldsymbol{q}^*}{\partial w_{ij}} = \frac{(\boldsymbol{q}^* - \boldsymbol{q}_d)^T}{E} \cdot \frac{\partial \boldsymbol{q}^*}{\partial w_{ij}} \tag{3.11}$$

$$= \frac{\mu}{\gamma E} \varphi(q_j) (\boldsymbol{q}^* - \boldsymbol{q}_d)^T \left[ \mathbb{I}_{4n} - \frac{\mu}{\gamma} W \cdot J_{\varphi}(\boldsymbol{q}) \right]^{-1} \boldsymbol{e}_i = \frac{\mu \varphi(q_j)}{\gamma E} \boldsymbol{\delta}^T S^{-1} \boldsymbol{e}_i. \tag{3.12}$$

which embodies the error term $\boldsymbol{\delta} = (\boldsymbol{q}^* - \boldsymbol{q}_d)^T$, the nonlinear activation contribution $\varphi(q_j^*)$, and the inverse of the sensitivity matrix $S = \mathbb{I}_{4n} - \frac{\mu}{\gamma} W \cdot J_{\varphi}(\boldsymbol{q})$. The biases $\boldsymbol{b}$ do not appear in the derivative because they are independent of $w_{ij}$. Denote the training iteration number by superscript $(\cdot)^{(k)}$ and learning rate by $\eta \in \mathbb{R}$. From above, we get the weight update scheme of strict gradient descent:

$$w_{ij}^{(k+1)} = w_{ij}^{(k)} - \eta \frac{\partial E}{\partial w_{ij}} = w_{ij}^{(k)} - \eta \frac{\mu \varphi(q_j)}{\gamma E} \boldsymbol{\delta}^T S^{-1} \boldsymbol{e}_i \tag{3.13}$$

**The application of GHR calculus.**   Strict gradient descent is effective for the learning process, but the quaternion structure of the weights matrix $W$ could not be preserved, where we should have every $4 \times 4$ block to be negative symmetric, except for the elements on the leading diagonal, like we have seen in the matrix expression of quaternionic multiplication by Eq. (2.1). GHR gradient descent over $\mathbb{H}$ can be solved from the following system. The first equation is the derivative of the loss function $E(\boldsymbol{q}) = |\boldsymbol{q} - \boldsymbol{q}_d|^2$ through Def. 2.2, where $\boldsymbol{q}$ is the stable state of the evolution equation and $\mathcal{I} = \{\boldsymbol{i}, \boldsymbol{j}, \boldsymbol{k}\}$. The second equation is by taking the differential of the sensitivity Eq. (3.8). For $\varrho \in \mathcal{J} = \{1, \boldsymbol{i}, \boldsymbol{j}, \boldsymbol{k}\}$, the following differential could be calculated by:

$$\frac{\partial E}{\partial \boldsymbol{q}^{\varrho}} = \frac{1}{4} \left[ \frac{\partial E}{\partial \boldsymbol{q}_s} - \sum_{\sigma \in \mathcal{I}} \frac{\partial E}{\partial \boldsymbol{q}_{\sigma}} \circ \sigma \right]^{\varrho} \tag{3.14}$$

$$\frac{\partial \boldsymbol{q}^{\varrho}}{\partial \boldsymbol{\omega}_{ij}^{\dagger}} = \frac{\mu}{\gamma} \left[ W^{\varrho} \circ \frac{\partial \boldsymbol{q}^{\varrho}}{\partial \boldsymbol{\omega}_{ij}^{\dagger}} + \frac{\partial W^{\varrho}}{\partial \boldsymbol{\omega}_{ij}^{\dagger}} \circ \boldsymbol{q}^{\varrho} \right] \tag{3.15}$$

where subscript $(\cdot)_{\varrho}$ is for quaternion components $\boldsymbol{q} = \boldsymbol{q}_s + \boldsymbol{q}_x \boldsymbol{i} + \boldsymbol{q}_y \boldsymbol{j} + \boldsymbol{q}_z \boldsymbol{k}$, and superscript is for quaternion involution introduced in Def. 2.1. This section shows that the quaternion structure-preserving descent exists, providing the foundation of the projection approach. But we would not apply this scheme for the algorithm. Formulas of quaternion calculus applied in this section could be found in [36], especially the generalized chain rule deriving Eq. (3.15). These lead to the steepest descent direction, which is the gradient of the loss function $E(\boldsymbol{q}^*)$:

$$\frac{\partial E}{\partial \boldsymbol{\omega}_{ij}^{\dagger}} = \sum_{\varrho \in \mathcal{J}} \frac{\partial E}{\partial (\boldsymbol{q}^*)^{\varrho}} \circ \frac{\partial (\boldsymbol{q}^*)^{\varrho}}{\partial \boldsymbol{\omega}_{ij}^{\dagger}}. \tag{3.16}$$

And the update of weights over $\mathbb{H}$ during the learning process under GHR calculus is thereby:

$$\boldsymbol{\omega}_{ij}^{(k+1)} = \boldsymbol{\omega}_{ij}^{(k)} - \eta \frac{\partial E}{\partial \boldsymbol{\omega}_{ij}^{\dagger}}. \tag{3.17}$$

**Periodic Projection Method.** As we have found in the above paragraph, direct gradient descent preserving the quaternionic structure by GHR calculus is too complicated and computationally costly. Therefore, we propose a strategy for the weights update named periodic projection. The main idea is to periodically do a Frobenius orthogonal projection with the following theorem every $K = 5\ to\ 10$ iterations of weights update in the training procedure. The proof of theorem is given in App. C.

**Theorem 3.3** (Frobenius orthogonal projection). *Let $\mathcal{L} = \text{span}\{L(1), L(\boldsymbol{i}), L(\boldsymbol{j}), L(\boldsymbol{k})\} \subset \mathbb{R}^{4\times 4}$ be the quaternion left multiplication submanifold with base Eq. (2.2). For any matrix $M \in \mathbb{R}^{4\times 4}$, define its orthogonal projection onto $\mathcal{L}$ under the Frobenius inner product as:*

$$\widetilde{M} = \sum_{\mu \in \{1, \boldsymbol{i}, \boldsymbol{j}, \boldsymbol{k}\}} c_\mu L(\mu) := \sum_{\mu \in \{1, \boldsymbol{i}, \boldsymbol{j}, \boldsymbol{k}\}} \frac{\langle M, L(\mu) \rangle_F}{\|L(\mu)\|_F^2} L(\mu). \tag{3.18}$$

*Then for any $A \in \mathcal{L}$, we have $\|M - A\|_F^2 \geq \|M - \widetilde{M}\|_F^2$, which means $\widetilde{M}$ is the closest approximation of $M$ in $\mathcal{L}$ under the meaning of the Frobenius measure.*

## 4  Experiments

**Algorithm and Configurations.** We built an experimental model of QSHNN with four quaternion neurons in Python and executed it on a PC with an Apple Silicon M1 chip, 16GB unified memory, aiming to verify the theoretical results including asymptotic stability, smoothness of trajectories, and the effectiveness of learning rules. The algorithm below gives the choice of hyper-parameters for the experiments and how the approaches in Sec. 3 are implemented, especially the difference made by the periodic projection method. Training results and characterization are shown in Fig $5 \sim 7b$.

---

**Algorithm 1:** Projection-based QSHNN learning algorithm

---

**Input:** Target $\boldsymbol{d} \in \mathbb{R}^{4N}$, where uniform distribution is obeyed by $\boldsymbol{d}_i \sim \mathcal{U}(-1, 1)$.
**Output:** Trained weight matrix $W \in \mathbb{R}^{4N \times 4N}$, Loss and accuracy evolution.
**Initialization:** Learning rate $\eta = 0.001 \sim 0.2$ with adaptive adjustment, projection period $\mathcal{P} = 10$, maximum epochs $T_{\max} = 30000$, error tolerance $\tau = 10^{-6}$, bias $b = 0.125 \sim 0.2$, network parameters $\mu = \gamma = 1$ by default, Weights initialization $w_{ij} \sim \mathcal{U}(-1, 1)$. Weights normalization by $W = W/(\varepsilon + \|W\|_\infty)$, where $\varepsilon = 10^{-12} \ll 0$ an arbitrary small number.

**for** $i = 1$ **to** $T_{\max}$ **do**                                           // main training loop
  Solve $\dot{\boldsymbol{q}} = -\gamma \boldsymbol{q} + \mu \boldsymbol{W} \boldsymbol{\varphi}(\boldsymbol{q}) + \mu \boldsymbol{b}$          // by Runge-Kutta numerical method
  $\boldsymbol{\delta} \leftarrow \boldsymbol{q}^* - \boldsymbol{d}$                                    // compute error vector
  **for** $i = 1, \ldots, N$ **do**
    **for** $j = 1, \ldots, N$ **do**
      $S = \mathbb{I}_{4n} - \frac{\mu}{\gamma} W \cdot J_\varphi(\boldsymbol{q})$                 // compute sensitivity matrix
      $w_{ij} \leftarrow w_{ij} - \eta \frac{\mu \varphi(q_j)}{\gamma} \boldsymbol{\delta}^T S^{-1} \boldsymbol{e}_i$        // strict gradient Eq.(3.13)

  **if** $i \equiv 0 \ (mod\ \mathcal{P})$ **then**                            // block-wise projection Eq.(3.18)
    $W \leftarrow \widetilde{W} = c_1 L(1) + c_i L(\boldsymbol{i}) + c_j L(\boldsymbol{j}) + c_k L(\boldsymbol{k})$
  **if** $\sum_n |\boldsymbol{\delta}^{(n)}|^2 < \tau$ **and** $\mathcal{P}|i$ **and** *accuracy=1.0* **then**          // stop criteria
    **break;**

---

Remove the projection process, we have the naive supervised learning Hopfield neural network (SHNN). It does not have the activation evolution in the quaternion region, whereby the Eq. (1.1). The training process and outcome for both QSHNN and SHNN are displayed in Fig $5 \sim 8$. Based on the periodic projection algorithm, we construct a prototype of robotic applications and demonstrate the effectiveness of the QSHNN in Fig. 8, where we also list the benchmark problems and baselines in App. E and F for the development of a complete module in robotic planning and control. They aim to embody the advantages of QSHNN in these important tasks.

**Fundamental Training Evaluation.** We design benchmark datasets by generating random sets of target quaternion states. Each target contains N quaternion-valued desired states $\boldsymbol{q}_d \in \mathbb{H}^N$, with each component sampled uniformly from the interval $[-1, 1]$. For each benchmark, we report three metrics: (i). Accuracy: the percentage of target sets for which the equilibrium state successfully converges

below the error threshold $\tau_1 = 10^{-6}$. More specifically, the proportion of quaternion components satisfying $q_i^* - d_i < \tau$ is defined by $accuracy \in [0, 1]$, which is included in the training stop criteria as $accuracy = 1.0$. (ii). Maximum iterations: the number of iterations allowed to conduct gradient descent and meet the convergence criteria on the randomly generated target set. $T_{max} = 30000$ for QSHNN, and $T_{max} = 10000$ for SHNN, which evaluates the effectiveness of the learning scheme. (iii). Equilibrium error: the neural system should be equipped with a unique equilibrium, and all randomly generated initial values can converge with an error less than $\tau_2 = 10^{-6}$, which is the indicator of asymptotic stability. Metric of error is mean square error (MSE) or Euclidean distance.

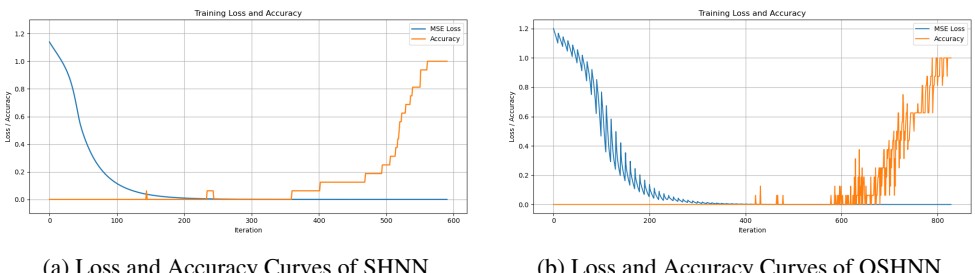

(a) Loss and Accuracy Curves of SHNN        (b) Loss and Accuracy Curves of QSHNN

Figure 5: (a). Loss and accuracy curves over training iterations of strict gradient descent Eq. (3.13). It implements a sufficient convergence rapidly but will not preserve the quaternionic structure, shown in Fig. 6a. (b). Loss and accuracy curved over training iterations with periodic projection Eq. (3.18), which perform good error reduction and preserve quaternionic structure simultaneously. Curves fluctuate more sharply since the projection disturb continuous gradient descent, shown in Fig. 6b~6d.

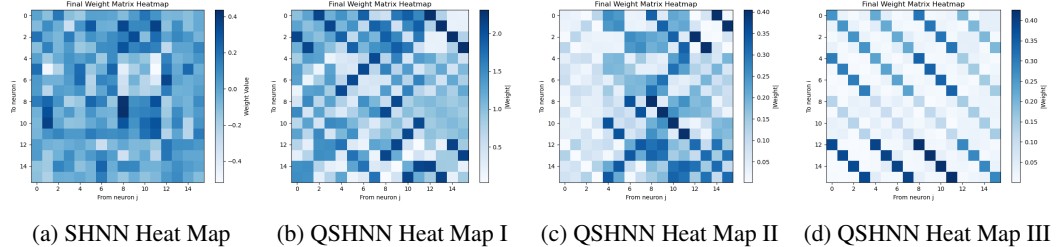

(a) SHNN Heat Map        (b) QSHNN Heat Map I        (c) QSHNN Heat Map II        (d) QSHNN Heat Map III

Figure 6: Heat map of network training outcome. Depth of the color indicates the magnitude of the absolute value of the weights. (a). Weights distribution trained by strict gradient descent Eq. (3.13). There are no obvious structure on blocks, thus this scheme is unable to keep quaternion correspondence by Eq. (2.1) on matrix. (b). Weights distribution trained by periodic projections Eq. (3.3). $4 \times 4$ blocks are quaternion symmetric matrix. The average deviation between the start configure and the target configure is $\sigma = 0.2$. (c). Weights distribution trained with periodic projections Eq. (3.3). We still have average deviation $\sigma = 0.2$, but with another random generation of initialization and target. (d). Weights distribution trained with periodic projections Eq. (3.3). Components of $ith$ neuron $\boldsymbol{q}^i$ have the same target, where $\boldsymbol{q}_s^i = \boldsymbol{q}_x^i = \boldsymbol{q}_y^i = \boldsymbol{q}_z^i$. Trained weights will concentrate on the main diagonal of the matrix. The model adapt its block-wise quaternionic structure to target-specific symmetries, a behaviour not apparent in the fully heterogeneous case shown in Fig. 6b and 6c.

## 5  Discussion

**Benchmark problems and Baselines.**    Based on the theoretical foundation of the proposed model, we plan to develop a complete application for robotic manipulator planning in the follow-up publication. Benchmark problems we consider are stated in App. E. Through these three levels of benchmarking, we can cover the widely recognized standard environment in the academic community and verify the versatility and scalability of direct sampling targets in industrial simulation, thereby fully demonstrating the combined advantages of QSHNN in terms of response speed, control accuracy, and online computing cost. Respectively, the baselines are the performance of current industrial algorithms, which are stated in App. F. In terms of the weaknesses mentioned of these baseline methods, QSHNN has the potential to outperform traditional strategies. Besides the guarantees on dynamical properties, powerful representation capability of QSHNN allows small-scale neural networks to complete tasks, thereby significantly improving the online planning reaction speed and reducing response time with the adjustment of the target.

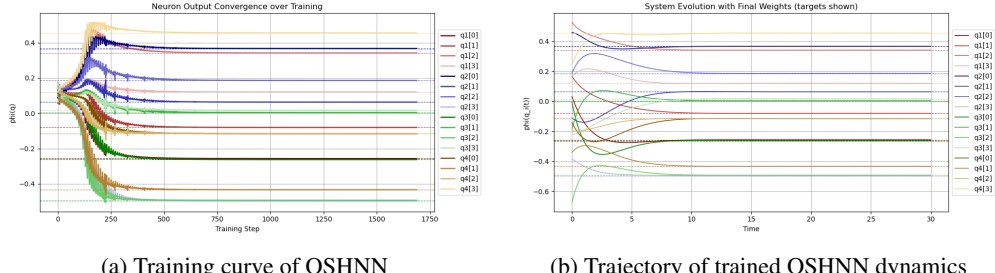

(a) Training curve of QSHNN        (b) Trajectory of trained QSHNN dynamics

Figure 7: (a). Training curve of equilibrium for the neural dynamical system governed by Eq. (3.2). Each color set represent the components of a quaternion neuron activation and converges to the target. (b). Evolution curve of the neural differential system governed by Eq. (3.2). Embody the asymptotical stability and smoothness of trajectories guaranteed by Thm. 3.7 and Thm. 3.6.

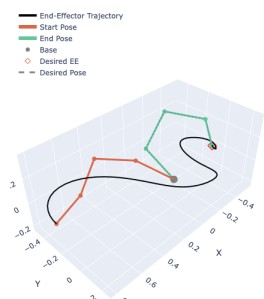

Figure 8: Robotic simulation with QSHNN as the path planning module. The scenario is aligned with Fig. 2. Here we conducted preliminary simulations in standard robotic environments PyBullet (generic physical simulation engine for robotics and control system in Python) to confirm that QSHNN can drive a robotic manipulator with four full-freedom joints from arbitrary initial joint configurations to a specified end-effector posture with the smoothness and convergence properties proven in the paper, we plan to develop the mature application in a follow-up publication with a complete evaluation of existing benchmarks App. E and comparison with baselines App. F.

**Comparative Analysis.** Regarding how QSHNN becomes an exclusive approach by its features with respect to traditional models, we make the following comparisons. (i): Supervised quaternion networks (QSNN), which only perform static regression from inputs to outputs without continuous-time stability guarantees or the ability to manipulate dynamical patterns. (ii): naïve quaternion-valued Hopfield neural network (QHNN) with a supervised readout, where the attractor dynamics are fixed after unsupervised weight initialization, so the supervised layer merely maps from pre-existing equilibrium states to outputs, and thus cannot modify the underlying attractor dynamics. (iii): proposed bio-inspired neural network (QSHNN), which generates continuous-time trajectories with provable global convergence and modifiable dynamics with physical information embedded.

**Limitations.** Throughout this research, we explore the potential to combine the memory-type Hopfield neural network with the mainstream method based on error propagation. The recent research on neuroscience revealed the surprising fact that the echo-location system of bats consists of only a small number of neurons as a core module [8], which motivates us to exploit the potential of bio-inspired recurrent neural networks. To implement this principle, we begin with the modification of continuous HNN and specify a concrete task in robotics by embedding physical information with quaternion. Though the theoretical foundation is established, only the development of the complete application with sufficient tests on the benchmark problems and comparison App. E with baselines App. F can make it more trustworthy, thereby giving us the confidence to further exploit the principle of general learning theory behind it, which is far more than the value of a single model.

## 6 Conclusions

We presented a quaternion-valued Hopfield-structured neural network that integrates structural consistency with smooth and stable learning dynamics. Modern HNN is mature in [23], there are also many hypercomplex-valued versions [12, 14, 32], but the plasticity of supervision is usually weak, and the global stability is insufficient, where our model fills the gap in exploration in this aspect. By combining supervised training with periodic projection, the model preserves quaternionic structure while achieving accurate and reliable convergence. The bounded curvature of trajectories ensures smooth evolution, and the network's design enables robust performance across randomly generated targets. Beyond orientation and control tasks, this approach also illustrates how embedding algebraic constraints into neural systems can yield both theoretical guarantees and practical benefits, offering a systematic path forward for structured learning in hypercomplex and non-commutative domains.

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

# Appendices

## A  Proof of Thm. 3.1: Existence and uniqueness of equilibrium

Firstly, the activation functions selected are usually bounded. For this reason, we define a constant upper bound for all of the activation functions appearing in the below defined as $\varphi \leq M$. We do not strictly demand that the activation function be uniformly continuous, but satisfying the Lipschitz condition. Out of a similar reason, we allocate each activation function a Lipschitz constant with respect to the subscript of notions as: $|\varphi_i(x) - \varphi_i(y)| \leq L_i |x - y|$.

To prove the existence of at least one critical point of the dynamic system 3.2, which is to say when we replace the variable of the differential with zero, and by definition, every solution point of the variable functions $v_i$ counts. Here is to say the below equation system is solvable.

$$v_i = \frac{\mu}{\gamma}\left[\sum_{j=1}^{n} w_{ij}\varphi_j(v_j) + b_i\right] \tag{A.1}$$

established for $x \in \mathbb{R}$, $i = 1, 2, \ldots n$. We denote the linear combination of symbols as matrix $A = \{\frac{\mu}{\gamma}w_{ij}\}_{n\times n}$ and bias or external current $\zeta = \{\frac{\mu}{\gamma}b_i\}_{n\times 1}$. And vector multiple value function $\varphi_i : \mathbb{R} \to \mathbb{R}$, define $\mathcal{F} := \{f | f : \mathbb{R} \to \mathbb{R}, \ f \ is \ Lipschitz \ continuous\}$, then $\boldsymbol{\varphi} \in \mathcal{F}^n \otimes \mathbb{R}^n$ is to be: $\boldsymbol{\varphi}(\boldsymbol{v}) = \{\varphi_i(v_i)\}_{n\times 1}$. The system governed by Eq. (3.2) can then be rewritten in matrix form:

$$\boldsymbol{v} = A\boldsymbol{\varphi}(\boldsymbol{v}) + \boldsymbol{\zeta} := \boldsymbol{F}(\boldsymbol{v}) \tag{A.2}$$

if we regard the right-hand side as a mapping $\boldsymbol{F}$ from $\mathbb{R}^n$ to $\mathbb{R}^n$, then the solution $\boldsymbol{v}$ can be treated as the fixed point of the mapping, allowing the analytic method in the proof.

$$\begin{aligned}
||\boldsymbol{F}(\boldsymbol{v})||^2 &= \sum_{i=1}^{n}\left[\frac{\mu}{\gamma}\left(\sum_{j=1}^{n} w_{ij}\varphi_j(v_j) + b_i\right)\right]^2 \\
&\leq \sum_{i=1}^{n}\frac{\mu^2}{\gamma^2}\left[\left(\sum_{j=1}^{n}|w_{ij}|M\right) + |I_i|\right]^2 \\
&:= \rho^2
\end{aligned} \tag{A.3}$$

From $\rho$ we form a bounded convex set $\Omega = \{\boldsymbol{v} : ||\boldsymbol{v}|| \leq \rho\}$, whereby Brouwer fixed point theorem applies to the proposition. Since $\boldsymbol{F}$ is a continuous mapping, there exists a $\boldsymbol{v^*} \in \Omega$ satisfying $\boldsymbol{F}(\boldsymbol{v^*}) = \boldsymbol{v^*}$, which implies the existence of the critical point in the system governed by Eq. (3.2).

The uniqueness of the system can be proved by inequality skills; the demand is converted to constraints of weights. Through reduction to absurdity, suppose there exist two different critical points $\boldsymbol{v}^*$, $\boldsymbol{u}^*$. Consider $\boldsymbol{L_1}$ norm of $v - u$ to be:

$$\begin{aligned}
||\boldsymbol{u}, \boldsymbol{v}||_1 &= \sum_{i=1}^{n}|u_i - v_i| = \sum_{i=1}^{n}\left|\frac{\mu}{\gamma}\sum_{j=1}^{n} w_{ij}[\varphi_j(u_j) - \varphi_j(v_j)]\right| \leq \sum_{i=1}^{n}\frac{\mu}{\gamma}\sum_{j=1}^{n}|w_{ij}| \cdot L_i \cdot |u_j - v_j| \\
&= \sum_{i=1}^{n}\sum_{j=1}^{n}\frac{\mu}{\gamma}L_i \cdot |w_{ij}| \cdot |u_j - v_j| = \sum_{j=1}^{n}\left(\sum_{i=1}^{n}\frac{\mu}{\gamma}L_i|w_{ij}|\right) \cdot |u_j - v_j| < \sum_{j=1}^{n}|u_i - v_i|
\end{aligned} \tag{A.4}$$

This inequality leads to a contradiction when the weights are restricted by

$$\sum_{i=1}^{n}\frac{\mu}{\gamma}L_i|w_{ij}| < 1 \tag{A.5}$$

With the above constraints on weights, simply put, the sum of weights cannot be too large; then the network operation has a certain single convergent point for bounded input. The essential conclusion provides us with a theoretical base for complex network behaviour design and learning rules, which have not been clearly stated and summarized in the previous articles. The research on Hopfield

neural network is relatively thorough, regardless of the stability or convergent pattern design. We transfer these basic theories and extend them to the construction of quaternion-valued network, in combination with knowledge of metric space and functional analysis.

For simplification, when we use the inequality A.5 for deeper deduction, we will suppose the coefficients in the front of weights to be a unit constant. And for the practical network operation, this condition will be satisfied during the procedure of weights normalization, where we set the norm of the weights matrix to be a constant and the significant information will not be lost.

# B    Proof of Thm. 3.2: Lyapunov stability criterion

We want to do some research on the stability of the quaternion-valued modified neural network. In the methodology Sec. 3, we will summarize the process to verify it has a semi-negative derivative function, concluding the asymptotic stability. $R_j$ and $C_j$ are the resistances of the neuron simulated by the electronic circuit. To judge the stability, we will be more specific here since there is a little possibility for a system to diverge to infinity, whereas merely being stable will not meet the whole requirement. The value of the output is expected to attach to a single point which, strictly speaking, is a critical point to be asymptotically stable, making the undesired situation such as chaos disappear.

$$V(t) = \frac{1}{2\gamma} \sum_{j=1}^{n} x_j^2 \tag{B.1}$$

By Brouwer's fixed point theorem, we conclude the existence of a critical point, meaning B.2 established for any $j$ from 1 to n. By calculating the difference between the sum of the $L2$ metric of two critical points, we conclude the uniqueness of the critical point, where:

$$\dot{\boldsymbol{x}} = F_j(t, \boldsymbol{x}) = 0 \tag{B.2}$$

Suppose $Ki$ is the Lipschitz constant of the function $\varphi_i(\boldsymbol{x})$, as the model of HNN is an autonomous system, we have $F_j(t, \boldsymbol{x}) = F_j(\boldsymbol{x})$ for every $j = 1, 2, ..., n$. Notice that:

$$
\begin{aligned}
\frac{\mathrm{d}V(t)}{\mathrm{d}t} &= \frac{1}{\gamma} \sum_{j=1}^{n} x_j \dot{x}_j \\
&= \frac{1}{\gamma} \sum_{j=1}^{n} x_j \left[ -\gamma x_j + \mu \sum_{i=1}^{n} w_{ji} \varphi_i(x_i) \right] \\
&\leq \sum_{j=1}^{n} \left( -x_j{}^2 + \frac{\mu}{\gamma} \sum_{i=1}^{n} |w_{ji}| L_i x_i x_j \right) \\
&\leq \sum_{j=1}^{n} \left[ -x_j{}^2 + \frac{\mu}{2\gamma} \sum_{i=1}^{n} |w_{ji}| L_i (x_i{}^2 + x_j{}^2) \right] \\
&= \sum_{j=1}^{n} \left[ -1 + \frac{\mu}{2\gamma} \sum_{i=1}^{n} (|w_{ji}| + |w_{ij}|) \right] x_j{}^2 \\
&\leq 0
\end{aligned}
\tag{B.3}
$$

By Lyapunov stability theory, the zero solution of the system is asymptotically stable such that the critical point of the original system is stable when the constraint of weights

$$\frac{\mu}{2\gamma} \sum_{i=1}^{n} (|w_{ji}| + |w_{ij}|) < 1 \tag{B.4}$$

applies. Throughout the whole procedure, we notice that the activating function does not need to possess smoothness or continuity, but satisfy the Lipschitz condition.

## C Proof of Thm. 3.3: Projection to quaternion left multiplication manifold

Let $\mathcal{L} = \mathrm{span}\{L(1), L(i), L(j), L(k)\} \subset \mathbb{R}^{4\times4}$. For any matrix $M \in \mathbb{R}^{4\times4}$, denote its orthogonal projection onto $\mathcal{L}$ under the Frobenius inner product as

$$\widehat{M} = \sum_{\mu \in \{1,i,j,k\}} \frac{\langle M, L(\mu)\rangle_F}{\|L(\mu)\|_F^2} L(\mu). \tag{C.1}$$

Then for any $A \in \mathcal{L}$, we have

$$\|M - A\|_F^2 \geq \|M - \widehat{M}\|_F^2. \tag{C.2}$$

**Proof**: Let $\mathcal{L} = \mathrm{span}\{L(1), L(i), L(j), L(k)\} \subset \mathbb{R}^{4\times4}$. We equip $\mathbb{R}^{4\times4}$ with the Frobenius inner product

$$\langle A, B\rangle_F = \mathrm{tr}(A^T B), \qquad \|A\|_F = \sqrt{\langle A, A\rangle_F}. \tag{C.3}$$

A direct calculation using the quaternion relations shows that for all $\mu, \nu \in \{1, i, j, k\}$,

$$\langle L(\mu), L(\nu)\rangle_F = \mathrm{tr}\big(L(\mu)^T L(\nu)\big) = 4\,\delta_{\mu\nu}, \tag{C.4}$$

so $\{L(1), L(i), L(j), L(k)\}$ is an orthogonal basis of $\mathcal{L}$ with $\|L(\mu)\|_F = 2$.

Now take any $M \in \mathbb{R}^{4\times4}$. By orthogonal decomposition, there exist unique coefficients $c_\mu$ and a residual $E \in \mathcal{L}^\perp$ such that

$$M = \sum_\mu c_\mu L(\mu) + E. \tag{C.5}$$

Taking the inner product of both sides with $L(\nu)$ gives

$$\langle M, L(\nu)\rangle_F = \sum_\mu c_\mu \langle L(\mu), L(\nu)\rangle_F = 4\,c_\nu, \tag{C.6}$$

hence,

$$c_\nu = \frac{1}{4}\langle M, L(\nu)\rangle_F = \frac{1}{4}\mathrm{tr}\big(M^T L(\nu)\big). \tag{C.7}$$

Define:

$$\widehat{M} = \sum_\mu \frac{\langle M, L(\mu)\rangle_F}{4} L(\mu) \tag{C.8}$$

to be the projection of $M$ onto $\mathcal{L}$. Then for any $A \in \mathcal{L}$, since $(M - \widehat{M}) \perp (\widehat{M} - A)$, the Pythagorean theorem yields

$$\|M - A\|_F^2 = \|M - \widehat{M}\|_F^2 + \|\widehat{M} - A\|_F^2 \geq \|M - \widehat{M}\|_F^2, \tag{C.9}$$

with equality if and only if $A = \widehat{M}$. This shows $\widehat{M}$ is the unique Frobenius-least-squares projection of $M$ onto $\mathcal{L}$.

## D Proof of smoothness: Curvature bound for the smoothness

Start at the equation of quaternion neurons,

$$\dot{q}_i = -q_i + f(z_i) := F_i(q_1, \ldots, q_n) \tag{D.1}$$

where we referred

$$\boldsymbol{z}_i := \sum_j W_{ij}\phi(\boldsymbol{q}_j) + \boldsymbol{b}_j$$

Thus the second derivative of quaternion variables is:

$$\ddot{q}_i = \sum_k \frac{\partial F_i}{\partial q_k} = \sum_k (\delta_{ik} + \frac{\partial f}{\partial z_i} W_{ik})\dot{q}_k \tag{D.2}$$

From where we could write the Jacobian matrix of the function $\ddot{\boldsymbol{q}}$ about the variable $\dot{\boldsymbol{q}}$ since there is no explicit appearance of any other variables and their relation is linear.

$$\ddot{\boldsymbol{q}} = J\dot{\boldsymbol{q}}, \;\; J := J_{\dot{\boldsymbol{q}}}(\ddot{\boldsymbol{q}}) \tag{D.3}$$

where:

$$J_{ik} = \frac{\partial \ddot{\boldsymbol{q}}_i}{\partial \dot{\boldsymbol{q}}_k} = -\delta_{ik} + \frac{\partial f}{\partial z_i} W_{ik} \tag{D.4}$$

Now we will estimate the norm of different parts in D.3

$$\begin{aligned}
||\dot{\boldsymbol{q}}||_\infty &= \max_i | -q_i + f(z_i)| \\
&\leq \max_i |q_i| + \max_i |f(z_i)| \\
&\leq 2
\end{aligned} \tag{D.5}$$

The last inequality step is because we set the activation function $f$ to be hyperbolic tangent, and for the quaternion variable. Consider the region surrounding the origin where the modulus is less than 1. For the Jacobian, we also have:

$$\begin{aligned}
||J||_\infty &= \max_i \sum_{k=1}^{n} \left| -\delta_{ik} + \frac{\partial f}{\partial z_i} W_{ik} \right| \\
&\leq 1 + \max_i \sum_{k=1}^{n} |W_{ik}| \\
&\leq 2
\end{aligned} \tag{D.6}$$

Combine D.6 and D.5 with D.3, we have a rough estimation of the 2nd derivative of $\boldsymbol{q}(t)$:

$$||\ddot{\boldsymbol{q}}||_\infty \leq ||J||_\infty \cdot ||\dot{\boldsymbol{q}}||_\infty \leq 2 \cdot 2 = 4 \tag{D.7}$$

Therefore, every component variable of $\boldsymbol{q}(t)$ has an upper bound of 4 on the second derivative. It gives a loose curvature bound on the trajectory in $\mathbb{R}^n$. In simpler terms, the path that q(t) traces through space is guaranteed to change direction smoothly. There are no sudden jerks, kinks, or high-frequency oscillations. The maximum bending or concavity of the trajectory is limited, which is important for smooth motion in control systems. If the actual engineering demands, we could get a more strict and smaller curvature $\kappa$ defined by [7]:

$$\kappa_i(t) = \frac{|\ddot{q}_i(t)|}{\left(1 + (\dot{q}_i(t))^2\right)^{3/2}} \leq ||\ddot{\boldsymbol{q}}||_\infty \leq 4 \tag{D.8}$$

which is no more than the second derivative of $q_i(t)$ whereby less than the estimated upper bound we deduced in this section, for any component with index $i$ in the trajectory of network evolution.

## E  Benchmark problems

I. **Robosuite PandaReach** [39] Robosuite's Franka Panda Reach mission only requires end-to-target position and pose alignment, which is closer to the real industrial scene. The environment has 6 degrees of freedom + claws, allowing us to focus on evaluating attitude control rather than grabbing strategies.

II. **OpenAI Gym FetchReach** [20] FetchReach is a 7-degree-of-freedom Fetch robotic arm target-alignment task. The target position is randomly generated in the environment (expandable to quaternions with attitude constraints), and the observation includes the relative position and direction of the end effector and the target, and the action directly gives the joint speed. In this environment, we can evaluate the response time, final error, and step time overhead of the QSHNN drive joint angle evolution to a specified quaternion pose.

III. **Random Workspace Target Alignment** [31] Build a UR5 or KUKA LBR model directly in MuJoCo, uniformly sample the attitude targets (including position and quaternion direction) in their workspace, and test them on a large number of random initial-target pairs.

## F    Baselines

I. **Damped Least-Squares Inverse Kinematics (IK) [6]**: A closed-loop control law that solves joint increments directly based on Jacobian matrices and can be run online in milliseconds. It ensures local asymptotic convergence, but it is strongly dependent on the initial point and often only achieves a low accuracy of rad.

II. **Covariant Hamiltonian Optimization for Motion Planning (CHOMP) [40]**: Optimized trajectory planning to generate smooth paths, but requires offline parameter adjustment and is time-consuming online.

III. **Stochastic Trajectory Optimization for Motion Planning (STOMP) [13]**: An iterative trajectory optimizer that samples noisy perturbations of an initial guess and weights them by smoothness and collision-avoidance costs; while it generates low-curvature paths, it depends heavily on the initial trajectory and requires tens to hundreds of milliseconds per planning episode, lacking global convergence guarantees.

IV. **RRT-Connect + B-Spline Smoothing [2]**: First, use a fast random tree (RRT-Connect) to quickly generate a feasible path, and then use B-Spline or quadratic programming to smooth the path. This not only retains the efficient connectivity of the sampling algorithm, but also obtains a certain degree of smoothing effect. However, the continuity and convergence lack strict guarantees.

## G    Absolute error to the target

The rapid convergence of exact error between the target state and the network state is already guaranteed by exponential or asymptotic stability under the framework of Lyapunov theory. Here we supplement a deduction for an explicit expression. Recall the notations for the trajectory and target are $\boldsymbol{q}(t) \in \mathbb{C}^2([0, +\infty])^n$, $\boldsymbol{q}_d \in \mathbb{R}^n$, and the square error is expressed by:

$$E(t) = \frac{1}{4\gamma} \|\boldsymbol{q}(t) - \boldsymbol{q}_d\|^2. \tag{G.1}$$

Take the time derivative of $E(t)$ by the chain rule:

$$\frac{dE(t)}{dt} = \frac{1}{\gamma}[\boldsymbol{q}(t) - \boldsymbol{q}_d] \cdot \frac{d}{dt}[\boldsymbol{q}(t) - \boldsymbol{q}_d] \tag{G.2}$$

$$= \frac{1}{\gamma}[\boldsymbol{q}(t) - \boldsymbol{q}_d]^T \cdot [-\gamma\mathbb{I} + \mu W J_\varphi(\boldsymbol{q})] \cdot [\boldsymbol{q}(t) - \boldsymbol{q}_d] \tag{G.3}$$

$$\leq -4\gamma E(t) + \frac{\mu}{\gamma}[\boldsymbol{q}(t) - \boldsymbol{q}_d]^T \cdot W \cdot [\boldsymbol{q}(t) - \boldsymbol{q}_d] \tag{G.4}$$

$$= \sum_{j=1}^{n} \left[ -1 + \frac{\mu}{2\gamma} \sum_{i=1}^{n} (|w_{ji}| + |w_{ij}|) \right] \cdot [q_j(t) - d_j]^2 \tag{G.5}$$

$$\leq -4\lambda E(t), \tag{G.6}$$

where the first equality simply comes from the evolution equation of QSHNN and the inequality comes from the direct computation in App. B, Eq. (B.3). The coefficient $\lambda$ is defined by:

$$\lambda := \gamma - \frac{\mu}{2} \sum_{i=1}^{n} (|w_{ji}| + |w_{ij}|) > 0. \tag{G.7}$$

The inequality above is guaranteed by Thm. 3.2. Hence, the error evolution over time satisfies:

$$\|q(t) - q_d\| \leq \|q(0) - q_d\|e^{-2\lambda t}. \tag{G.8}$$

Thus, the exact distance between the network state and the target will exponentially descend to infinitesimal ($E \ll 1$) and be negligible in practical operation. For the error between the equilibrium of the QSHNN and the target, which is the loss function of the training process, the model is already verified on a large enough set of targets, and the training ends consistently with the criterion that the mean square error is less than $10^{-6}$, which is stated in Para. 4 of Sec. 4.

