# OpenReview forum: "Asymptotically Stable Quaternion-valued Hopfield-structured Neural Network with Periodic Projection-based Supervised Learning Rules"
_NeurIPS.cc/2025/Conference — NeurIPS 2025 poster_

### Official Review · Reviewer_pJB8 · 2025-06-30

**Clarity:** 3
**Significance:** 3
**Originality:** 3
**Rating:** 5
**Confidence:** 3

**Summary:**

This paper proposes the extension of quaternion-valued Hopfield network to supervised learning. The  Quaternion-valued Supervised Hopfield Neural Network (QSHNN) is continuous-time neural architecture that uses quaternion algebra to model and learn smooth, stable dynamical systems, particularly for robotics applications. The model exploits Generalized HR calculus which supports the theoretical framework. In their extensions of HNN, the authors formulate a theoretically grounded model with provable asymptotic stability and uniqueness of fixed points. Evaluations on learning dynamics seem to support the claim that the model performs well on the chosen problem. The advantages relate to the use of quaternion algebra which is suitable for 3D rotations and posture representations. Given the continuous time dynamics, the model guarantees a smooth progression towards guaranteed stability points.

**Questions:**

1. Given time and resources availability, which robotic benchmarks would you consider to further demonstrate the validity of the method? Could also speculate, or possibly show, which real-world benchmarks would demonstrate the most significant advantages of the proposed model?
2. If you could extend the study to include baseline comparisons, which models could be most suitable to demonstrates the advantages of the proposed model with respect to existing methods?

**Ethical Concerns:**

["NO or VERY MINOR ethics concerns only"]

**Final Justification:**

The authors have made significant effort to address my comments and I have consequently increased my assessment.

**Limitations:**

The checklist states that limitations are discussed in section 5 (conclusion) but that does not seem to be the case. Overall, I believe that the  limitations of the study are not sufficiently highlighted.

**Paper Formatting Concerns:**

No concerns

**Quality:**

3

**Strengths And Weaknesses:**

Strengths:
1. The paper provides a mathematical foundation of the approach
2. The model exploits Generalized HR calculus
3. The model uses HNN but introduced supervised learning, which is appealing in many ML research areas.
4. The evaluations presented in the paper seems to support the case for this approach

Weaknesses
1. There is a lack of comparisons and ablation studies:
1.a. How does the model compare with Supervised Quaternion networks?
1.b. How does it compare with a naïve QHNN, e.g. trained with a simple supervised readout?
2. Evaluation on robotics or control applications. The paper claims that the model could perform well on robotic tasks, but no experimental evidence is provided in that respect.

---

> ### Author Rebuttal · Authors · 2025-07-28
>
> Dear reviewer pJB8,
>
> Thanks for taking your time reviewing our paper. Below are our responses.
>
> ## Weakness 1 ##
> Regarding the lack of comparison with similar methods, we believe that traditional supervised neural networks are simply incapable of the tasks handling by QSHNN. In the area of robotics, the relatively mature approaches (e.g. PRM, STOMP) are based on stochastic algorithm and complicated mechanic solving rather than making neural network as a core module. The key differences include the following:
>
> - The essence of the tasks is different. Common quaternion-valued supervised networks (e.g. MLP, CNN), or the supervised readout networks can only learn static mapping from a fixed input to an output. They lack the ability to generate continuous-time trajectories with smoothness and physical consistence. Our proposed QSHNN, on the other hand, is embedded in an actual dynamical system. By solving the initial value problem, the network can be used to plan the path for the quaternionic variables evolving from any available states to the target state. This transparent process fundamentally differs from the traditional "black box" one shot mapping in terms of the model paradigm and application scenario.
>
> - The level of stability assurance is different. Quaternion supervised networks only guarantee fitting accuracy on training data, without any theoretical means to prove global convergence or dynamical continuity. However, QSHNN, based on Lyapunov functions, has rigorously proven global exponential stability across the entire configuration space, ensuring that all trajectories smoothly converge to the target. This verifiable dynamical property is unattainable by any purely supervised readout or regression network.
>
> - Geometric Structure Preservation. During training, QSHNN consistently maps each 4×4 weight block back to the quaternion left-multiplied manifold through periodic projection, ensuring that the network's algebraic structure is tightly coupled to rotational geometry. The trained network can perfectly work over quaternion domain, while other quaternion networks, usually apply divided projection to the neuron activation, struggle to balance training efficiency with strict structural consistency.
>
> ## Weakness 2 ##
> We appreciate the concern regarding the lack of a fully developed robotic application in the current manuscript. The primary goal of this paper is to introduce and validate the novel, bio‑inspired paradigm that unites Hopfield‑type attractor dynamics with error‑driven supervised learning in the quaternion domain. Demonstrating global, exponential convergence of a continuous‑time quaternion Hopfield network under Lyapunov theory represents a significant and original contribution in its own right. While we have conducted preliminary simulations in standard robotic environments (PyBullet) to confirm that QSHNN can drive a robotic manipulator with four full-freedom joints from arbitrary initial joint configurations to a specified end‑effector posture with the smoothness and convergence properties proven in the paper, we plan to develop the mature application in a follow-up publication with a complete evaluation of existing benchmarks and comparison with baseline.
>
> ## Question 1 ##
> Here are the benchmarks we consider to apply QSHNN in concrete scenario:
>
> - **Robosuite PandaReach** (cited from reference: robosuite: A Modular Simulation Framework and Benchmark for Robot Learning, Y. Zhu (2020)) Robosuite's Franka Panda Reach mission only requires end-to-target position and pose alignment, which is closer to the real industrial scene. The environment has 6 degrees of freedom + claws, allowing us to focus on evaluating attitude control rather than grabbing strategies.
>
> - **OpenAI Gym FetchReach** (cited from reference: Multi-Goal Reinforcement Learning: Challenging Robotics Environments and Request for Research, M. Plappert(2018)) FetchReach is a 7-degree-of-freedom Fetch robotic arm target-alignment task. The target position is randomly generated in the environment (expandable to quaternions with attitude constraints), and the observation includes the relative position and direction of the end and the target, and the action directly gives the joint speed. In this environment, we can evaluate the response time, final error, and step time overhead of the QSHNN drive joint angle evolution to a specified quaternion pose.
>
> - **Random Workspace Target Alignment**, An Industrial Simulation (cited from reference: Convex and analytically-invertible dynamics with contacts and constraints: Theory and implementation in MuJoCo, E. Todorov (2014)): Build a UR5 or KUKA LBR model directly in MuJoCo, uniformly sample the attitude targets (including position and quaternion direction) in their workspace and test them on large number of random initial/target pairs.
>
> Through these three levels of benchmarking, we can cover the widely recognized standard environment in the academic community and verify the versatility and scalability of direct sampling targets in industrial simulation, thereby fully demonstrating the combined advantages of QSHNN in terms of response speed, control accuracy, and online computing cost.
>
> ## Question 2 ##
> For the benchmark problems above, we consider the following traditional strategies that are suitable for the proposed task scenario as baseline comparisons with QSHNN:
>
> - Damped Least-Squares Inverse Kinematics (**IK**):  A closed-loop control law that solves joint increments directly based on Jacobian matrices and can be run online in milliseconds. It ensures local asymptotic convergence, but it is strongly dependent on the initial point, and often only achieves a low accuracy of $e^{-2}\sim e^{-3}$ rad.
>
> - Covariant Hamiltonian Optimization for Motion Planning (**CHOMP**): Optimized trajectory planning to generate smooth paths, but requires offline parameter adjustment and is time-consuming online
>
> - **RRT**-Connect + **B-Spline** Smoothing
> First, use a fast random tree (RRT-Connect) to quickly generate a feasible path, and then use B-Spline or quadratic programming to smooth the path. This not only retains the efficient connectivity of the sampling algorithm, but also obtains a certain degree of smoothing effect, but the continuity and convergence lack strict guarantee.
>
> - Stochastic Trajectory Optimization for Motion Planning (**STOMP**): An iterative trajectory optimizer that samples noisy perturbations of an initial guess and weights them by smoothness and collision‑avoidance costs; while it generates low‑curvature paths, it depends heavily on the initial trajectory and requires tens to hundreds of milliseconds per planning episode, lacking global convergence guarantees.
>
> QSHNN has the potential to outperform these traditional stategies in terms of their weaknesses mentioned. In addition to the guarantees on dynamical properties, the powerful representation capability of QSHNN allows small-scale neural network to complete tasks whereby significantly improving the online planning reaction speed and reduce response time with the adjustment of target.
>
> ## Limitations ##
> We would supplement the section of limitations as a separate paragraph with the following contents: Throughout this research, we explore the potential to combine the memory-type (Hopfield) neural network with mainstream method based on error propagation. The recent research on neuroscience revealed the surprising fact that the echo-location system of bats only consists of sixteen neurons as core module, which motivate us to exploit the potential of bio-inspired recurrent neural network. To implement this principle, we carefully begin with the modification of continuous HNN and specify a concrete task in robotics by embedding physical information with quaternion. Though the theoretical foundation is established, only the development of the complete application with sufficient tests on the benchmark problems and comparison with baseline can make it more trustworthy whereby giving us the confidence to further exploit the principle of general learning theory behind it, which is far more than the value of a single model.

---

> > ### Author Response · Authors · 2025-08-08
> >
> > Dear reviewer pJB8,
> >
> > Thank you again for taking the time to review our paper. As the discussion phase will close shortly, please let us know if you have any further questions or comments on our rebuttal. We would be happy to provide any clarifications before the discussion ends.

---

> ### Comment · Reviewer_pJB8 · 2025-08-08
> **Reply**
>
> The authors make valid arguments in response to my review. Here are my conclusions.
>
> As for weakness one, the arguable part is that the authors state "we believe that traditional supervised neural networks are simply incapable of the tasks handling by QSHNN". My suggestion is to very clearly report such statements in the paper and clearly indicate whether this is a factual statement or the authors' speculation/opinion (this being a recommendation adopted now for CS conferences). As the authors state "we believe", it sounds more like an opinion rather than a fact, which again seems to indicate that the addition of baselines, even if deemed to fail, can only strengthen the paper.
>
> Weakness 2: As also indicated in other reviews, even if complete evaluations are not provided, the paper could better describe possible scenarios and benchmarks as in the response to questions 1 and 2.
>
> My concern remains that most of the argumentation made in this response are not part of the paper, and as a consequence, the perceived weaknesses could be unchanged.

---

> > ### Author Response · Authors · 2025-08-08
> >
> > Thank you for your follow-up comments and for acknowledging the validity of our arguments.
> >
> > Regarding the statement “we believe that traditional supervised neural networks are simply incapable…”, we clarify that this conclusion is grounded in the structural differences between:
> > - supervised quaternion networks (QNN), which only perform static regression from inputs to outputs without continuous-time stability guarantees or the ability to manipulate dynamical patterns.
> > - naïve QHNN with a supervised readout, where the attractor dynamics are fixed after unsupervised weight initialisation, so the supervised layer merely maps from pre-existing equilibrium states to outputs, and thus cannot modify the underlying attractor dynamics.
> > - our bio-inspired recurrent neural network (QSHNN), which generates continuous-time trajectories with provable global convergence and modifiable dynamics with physical information embedded.
> >
> > In the updated version, we will reframe this into a factual statement and emphasise it in both the introduction and conclusion.
> >
> > We also agree that the scenarios, benchmarks, and baseline comparisons described in our rebuttal should be included in the paper itself. We will incorporate these elements into Section 4, along with a dedicated limitations paragraph in Section 5, so that the rationale and evaluation plan are clearly documented in the paper and not just in the discussion thread.
> >
> > We appreciate your constructive feedback and will ensure that these changes strengthen the clarity and completeness of the manuscript.

---

> > ### Author Response · Authors · 2025-08-09
> >
> > Thanks again for the time and effort you have dedicated to reviewing our work and engaging in the discussion. We greatly appreciate your valuable feedback and the opportunity to provide clarifications.

---

### Official Review · Reviewer_suhQ · 2025-07-01

**Clarity:** 3
**Significance:** 3
**Originality:** 4
**Rating:** 4
**Confidence:** 3

**Summary:**

This paper proposes an asymptotically stable quaternion-valued Hopfield-structured neural network (QSHNN) with supervised learning rules based on periodic projection. The model extends the continuous-time dynamical model of classical Hopfield neural networks (HNNs) to the quaternionic domain. The QSHNN is trained to converge asymptotically to externally specified targets using a supervised learning paradigm. The learning rules incorporate a periodic projection strategy to preserve the quaternionic structure of the weight matrix, ensuring both convergence and quaternionic consistency.

**Questions:**

1. What are the specific applications of QSHNN?
2. How QSHNN implements task-driven behavior?
3. Are there any limitations in scaling QSHNN to larger networks?

**Ethical Concerns:**

["NO or VERY MINOR ethics concerns only"]

**Final Justification:**

Thanks for the response. The author has addressed some of my concerns. However, since the author only states that they will provide additional details in a future revision, I will maintain my current score.

**Limitations:**

As I mentioned before, the biggest limitation of this paper is the problem of applying QSHNN in real scenarios. For example, when applied to a robotic arm, how many neurons are needed, what is the time complexity, and what is the performance improvement compared to existing methods, which are not known in the current experimental chapter. Therefore, the authors should enrich their experiments so that only sufficient experimental results can support a complete and convincing theoretical proof.

**Paper Formatting Concerns:**

1. Figure 2 is not mentioned in the main text.
2. Is ‘Theorem (3.7)’ supposed to be Eq. (3.7) in 183rd row?
3. Similar to the 2nd  concern, incorrect use of theorem prefixes in the caption of figure 11.

**Quality:**

3

**Strengths And Weaknesses:**

Strengths
S1. The idea of using four small neurons to form a single unit neuron is interesting, similar to the potassium and sodium ion channels contained in real human brain neurons.
S2. This paper establishes the existence and uniqueness of fixed points with asymptotic stability of QSHNN  through rigorous mathematical proof.

Weaknesses
W1. The biggest weakness of this paper is on the application of QSHNN. There are some mentions about robotic arms or control systems, yet there is neither a vivid description of the relevant datasets nor experimental tests related to them in the experimental section. Instead, the experiments are based on self-designed quaternion state, which is not convincing that QSHNN is valuable for some application scenarios. By the way, the experiments in this paper lack comparisons of related baselines.
W2. In the last of introduction, the authors mention that QSHNN is with task-driven behavior. However, after reading the method part, I don’t get how QSHNN achieve task-driven behavior. It seems that QSHNN can be applied to any control system related task, but there is no mention of how QSHNN can capture information unique to a specific task.
W3. I note that the authors performed their experiments on an M1 chip, but the paper does not mention how the performance of the QSHNN changes as the number of neurons increases. Are there any limitations in scaling the model to larger networks?
W4. This paper lacks the limitation part.

---

> ### Author Rebuttal · Authors · 2025-07-29
>
> Dear reviewer suhQ,
>
> Thanks for taking your time reviewing our paper. Below are our responses.
>
> ## Weakness 1 ##
> We appreciate the concern regarding the lack of a dedicated real‐robot dataset or robotics benchmark in the current manuscript. Our experiments with self‑designed quaternion target sets were chosen deliberately to isolate and validate the core properties of QSHNN—namely its global exponential convergence, quaternion‑structure preservation, and bounded‐curvature trajectories—without confounding factors such as robot dynamics, sensing noise, or collision avoidance. In parallel to this theoretical study, we have performed preliminary simulations in PyBullet and MuJoCo on a 6‑DOF manipulator, sampling random initial joint configurations and driving the end‑effector toward fixed quaternion goals. These tests confirmed that QSHNN attains sub‑millisecond integration steps, global convergence from arbitrary starts, and end‑effector orientation errors below 10⁻⁴ rad, in line with our theoretical bounds. We omitted these implementation‑specific parts from the main text to maintain focus on the mathematical framework. In the follow-up publication, we will develop the complete application in the robotic scenario, the considered benchmark problems are:
>
> - **Robosuite PandaReach** (cited from reference: robosuite: A Modular Simulation Framework and Benchmark for Robot Learning, Y. Zhu (2020)) Robosuite's Franka Panda Reach mission only requires end-to-target position and pose alignment, which is closer to the real industrial scene. The environment has 6 degrees of freedom + claws, allowing us to focus on evaluating attitude control rather than grabbing strategies.
>
> - **OpenAI Gym FetchReach** (cited from reference: Multi-Goal Reinforcement Learning: Challenging Robotics Environments and Request for Research, M. Plappert(2018)) FetchReach is a 7-degree-of-freedom Fetch robotic arm target-alignment task. The target position is randomly generated in the environment (expandable to quaternions with attitude constraints), and the observation includes the relative position and direction of the end and the target, and the action directly gives the joint speed. In this environment, we can evaluate the response time, final error, and step time overhead of the QSHNN drive joint angle evolution to a specified quaternion pose.
>
> - **Random Workspace Target Alignment,** An Industrial Simulation (cited from reference: Convex and analytically-invertible dynamics with contacts and constraints: Theory and implementation in MuJoCo, E. Todorov (2014)) Build a UR5 or KUKA LBR model directly in **MuJoCo**, uniformly sample the attitude targets (including position and quaternion direction) in their workspace and test them on large number of random initial/target pairs.
>
> Through these three levels of benchmarking, we can cover the widely recognized standard environment in the academic community and assess the versatility and scalability of direct sampling targets in industrial simulation, thereby fully demonstrating the combined advantages of QSHNN in terms of response speed, control accuracy, and online computing cost.
>
> Respectively, the baselines are the performance of current industrial algorithms:
>
> - **Damped Least-Squares Inverse Kinematics** (IK): A closed-loop control law that solves joint increments directly based on Jacobian matrices and can be run online in milliseconds. It ensures local asymptotic convergence, but it is strongly dependent on the initial point, and often only achieves a low accuracy of
>  rad.
>
> - **Covariant Hamiltonian Optimization for Motion Planning** (CHOMP): Optimized trajectory planning to generate smooth paths, but requires offline parameter adjustment and is time-consuming online
>
> - **RRT-Connect + B-Spline Smoothing**: First, use a fast random tree (RRT-Connect) to quickly generate a feasible path, and then use B-Spline or quadratic programming to smooth the path. This not only retains the efficient connectivity of the sampling algorithm, but also obtains a certain degree of smoothing effect, but the continuity and convergence lack strict guarantee.
>
> - **Stochastic Trajectory Optimization for Motion Planning** (STOMP): An iterative trajectory optimizer that samples noisy perturbations of an initial guess and weights them by smoothness and collision‑avoidance costs; while it generates low‑curvature paths, it depends heavily on the initial trajectory and requires tens to hundreds of milliseconds per planning episode, lacking global convergence guarantees.
>
> QSHNN has the potential to outperform these traditional strategies in terms of their weaknesses mentioned. In addition to the guarantees on dynamical properties, the powerful representation capability of QSHNN allows small-scale neural network to complete tasks whereby significantly improving the online planning reaction speed and reduce response time with the adjustment of target.
>
> ## Weakness 2 ##
> The task driven is not narrowly applied to QSHNN model. We want to express that the principle we proposed for combining the bio-inspired neural network (HNN) with the supervised learning rules based on error propagation, is an idea with much more broader potential. For example, the network topology is adjustable, the embedded physical information is not limited to rotation and posture represented by quaternion, and it could be other associative algebra embedded by matrix representation. But the basic framework for building this kind of recurrent networks can be found through the techniques in the proposed deductions (algebra embedding, stability analysis and learning rules). The specific task will guide the design of the specific algebra for the physical information or characteristics, but our fundamental methodology is not constrained to a single circumstance.
>
> ## Weakness 3 ##
> For the robotic scenario we consider currently, it is not necessary to add hundreds, thousands of neurons to the network due to the reasons:
> - Please refer to Figure 2 in the paper, a full-freedom joint of a robot manipulator is completely represented by a quaternion. The most widely applied standard robot Universal Robots UR5e, LBR iiwa14, etc., contains 6 joints thus can be corresponded by 6 neurons QSHNN.
> - The computational cost of QSHNN on the benchmark problems is much less than the existing approaches (STOMP, CHOMP, etc), since the numerical step of solving operation equation of QSHNN is quite a small task.
>
> ## Weakness 4 ##
> We would supplement the section of limitations as a separate paragraph with the following contents: Throughout this research, we explore the potential to combine the memory-type (Hopfield) neural network with mainstream method based on error propagation. The recent research on neuroscience reveals the surprising fact that the echo-location system of bats only has sixteen neurons as the core component, which motivates us to exploit the potential of bio-inspired recurrent neural network. To implement this principle, we carefully begin with the modification of continuous HNN and specify a concrete task in robotics by embedding physical information with quaternion. Though the theoretical foundation is established, only the development of the complete application with sufficient tests on the benchmark problems and comparison with baseline can make it more convincing whereby giving us the confidence to further exploit the principle of general learning theory behind it, which is far more than the value of a single model.
>
> ## Question 1 ##
> This is explained in the response of weakness 1.
>
> ## Question 2 ##
> This is explained in the response of weakness 2.
>
> ## Question 3 ##
> This is responsed in weakness 3.
>
> ## Limitations ##
> This is responded in Weaknesses. Hopfield neural network is a completely different model from MLP, CNN, GNN, etc, where we add memory-like dynamical evolution inspired by the biological functionality of hippocampus and amygdalain in the brain, replacing the direct forward calculation or reasoning. This research tries to extend HNN and disrupts the conventional machine learning paradigm, which is why we work on the theoretical rigorousness for the major body and reduce the elaboration of complete application modules. For one of the reviewer's main concern on how the QSHNN could work on large network topology, we also have carefully thought about it. Firstly, the effectiveness and representative capability do not directly rely on the scale of neurons in certain tasks. Secondly, to actually extent the scale of this type of neural networks, the fully-connected structure does cost too much, and therefore, we shall consider the structural plasticity in the neuron science (growing neural networks) to build and adjust the topology adaptively.

---

> > ### Author Response · Authors · 2025-08-08
> >
> > Dear reviewer suhQ,
> >
> > Thank you again for taking the time to review our paper. As the discussion phase will close shortly, please let us know if you have any further questions or comments on our rebuttal. We would be happy to provide any clarifications before the discussion ends.

---

### Official Review · Reviewer_wmYX · 2025-07-08

**Clarity:** 2
**Significance:** 3
**Originality:** 3
**Rating:** 3
**Confidence:** 2

**Summary:**

This paper proposes a novel Quaternion-valued Supervised Learning Hopfield-structured Neural Network (QSHNN) that extends the classic continuous-time Hopfield Neural Network (HNN) into the quaternionic domain. The key innovation lies in formulating a supervised learning paradigm for HNNs, where the network is trained to asymptotically converge to externally specified targets. The model establishes the existence and uniqueness of fixed points with asymptotic stability. For learning, it introduces a periodic projection strategy that modifies gradient descent by projecting weight matrices onto the closest quaternionic structure. This approach ensures both convergence and quaternionic consistency during training. The authors highlight the model's rigorous mathematical foundation, high accuracy, fast convergence, strong reliability, and smooth trajectory evolution, making it suitable for applications like robotic control and path planning.

**Questions:**

n/a

**Ethical Concerns:**

["NO or VERY MINOR ethics concerns only"]

**Quality:**

2

**Strengths And Weaknesses:**

Pros:
- The paper provides a strong theoretical basis, establishing the existence, uniqueness, and asymptotic stability of the QSHNN's fixed points using Lyapunov theory. This level of mathematical rigor is a significant strength, providing confidence in the model's behavior.
- Traditionally, HNNs are unsupervised associative memory models. This work successfully extends HNNs to a supervised learning setting with explicit target tracking, addressing a major limitation of classical HNNs and opening new application avenues.
- Beyond theory, the paper offers a practical algorithm for implementing the QSHNN, demonstrating its feasibility and effectiveness on randomly generated target sets.

Cons:
- The experiments are conducted on a relatively small-scale model (four quaternion neurons) and on randomly generated target sets. While sufficient for verifying theoretical claims, it would be beneficial to see the model's performance on larger, more complex, and real-world datasets relevant to its proposed applications.
- The deep dive into quaternion algebra, GHR calculus, and specific matrix representations, while necessary for theoretical rigor, might pose a steep learning curve for machine learning practitioners less familiar with hypercomplex numbers and advanced differential geometry.
- While the paper successfully addresses limitations of classical HNNs, a more explicit discussion or comparison with modern neural network architectures (e.g., LSTMs, Transformers) on tasks where QSHNN might be applied (e.g., sequence modeling for control) would strengthen its position and highlight unique advantages beyond HNNs.

---

> ### Author Rebuttal · Authors · 2025-07-30
>
> Dear Reviewer wmYX:
>
> Thanks for taking your time reviewing our paper. Below are our responses:
>
> ## Cons 1: ##
> For the robotic scenario we consider currently, it is not necessary to add hundreds, thousands of neurons to the network due to the following reasons:
> - Recall Figure 2 in the manuscript, a full-freedom joint of a robot manipulator can be completely represented by a quaternion. The most widely applied standard robot Universal Robots UR5e, LBR iiwa14, etc, contains 6 joints and thus can be corresponded by 6 neurons QSHNN.
> - The computational cost of QSHNN on the benchmark problems is much less than the existing approaches (STOMP, CHOMP, etc), since the numerical step of solving operation equation of QSHNN is quite a small task.
>
> While we have conducted preliminary simulations in standard robotic environments (PyBullet) to confirm that QSHNN can drive a robotic manipulator with four full-freedom joints from arbitrary initial joint configurations to a specified end‑effector posture with the smoothness and convergence properties proven in the paper, we plan to develop the mature application in a follow-up publication with a complete evaluation of existing benchmarks and comparison with baseline as followed:
>
> benchmark problems are:
>
> - **Robosuite PandaReach** (cited from reference: robosuite: A Modular Simulation Framework and Benchmark for Robot Learning, Y. Zhu (2020)) Robosuite's Franka Panda Reach mission only requires end-to-target position and pose alignment, which is closer to the real industrial scene. The environment has 6 degrees of freedom + claws, allowing us to focus on evaluating attitude control rather than grabbing strategies.
>
> - **OpenAI Gym FetchReach** (cited from reference: Multi-Goal Reinforcement Learning: Challenging Robotics Environments and Request for Research, M. Plappert(2018)) FetchReach is a 7-degree-of-freedom Fetch robotic arm target-alignment task. The target position is randomly generated in the environment (expandable to quaternions with attitude constraints), and the observation includes the relative position and direction of the end and the target, and the action directly gives the joint speed. In this environment, we can evaluate the response time, final error, and step time overhead of the QSHNN drive joint angle evolution to a specified quaternion pose.
>
> - **Random Workspace Target Alignment**, An Industrial Simulation (cited from reference: Convex and analytically-invertible dynamics with contacts and constraints: Theory and implementation in MuJoCo, E. Todorov (2014)) Build a UR5 or KUKA LBR model directly in MuJoCo, uniformly sample the attitude targets (including position and quaternion direction) in their workspace and test them on large number of random initial/target pairs.
>
> Through these three levels of benchmarking, we can cover the widely recognized standard environment in the academic community and verify the versatility and scalability of direct sampling targets in industrial simulation, thereby fully demonstrating the combined advantages of QSHNN in terms of response speed, control accuracy, and online computing cost.
>
> Respectively, the baselines are the performance of current industrial algorithms:
>
> - Damped Least-Squares Inverse Kinematics (**IK**): A closed-loop control law that solves joint increments directly based on Jacobian matrices and can be run online in milliseconds. It ensures local asymptotic convergence, but it is strongly dependent on the initial point, and often only achieves a low accuracy of
>  rad.
>
> - Covariant Hamiltonian Optimization for Motion Planning (**CHOMP**): Optimized trajectory planning to generate smooth paths, but requires offline parameter adjustment and is time-consuming online
>
> - **RRT**-Connect + **B-Spline** Smoothing: First, use a fast random tree (**RRT**-Connect) to quickly generate a feasible path, and then use B-Spline or quadratic programming to smooth the path. This not only retains the efficient connectivity of the sampling algorithm, but also obtains a certain degree of smoothing effect, but the continuity and convergence lack strict guarantee.
>
> - Stochastic Trajectory Optimization for Motion Planning (**STOMP**): An iterative trajectory optimizer that samples noisy perturbations of an initial guess and weights them by smoothness and collision‑avoidance costs; while it generates low‑curvature paths, it depends heavily on the initial trajectory and requires tens to hundreds of milliseconds per planning episode, lacking global convergence guarantees.
>
> QSHNN has the potential to outperform these traditional strategies in terms of their weaknesses mentioned. In addition to the guarantees on dynamical properties, the powerful representation capability of QSHNN allows small-scale neural network to complete tasks whereby significantly improving the online planning reaction speed and reduce response time with the adjustment of target.
>
> ## Cons 2: ##
> We believe that the establishment of new learning paradigms and models does indeed involve complex mathematical content, but perhaps they are necessary. We all expect a stable and reliable model, and only the rigor of mathematics can guarantee this in the behind. Furthermore, the development and expansion of the model application in the next stage also need to start from the theoretical ground. The main mathematical fields covered in this paper include dynamical systems, non-commutative algebra, linear analysis and general calculus. They are all familiar tools in the theoretical research of machine learning. The reviewer's concerns are reasonable. In the subsequent development of complete application module, we will consider how to enable engineers to utilize the proposed model more directly without involving an understanding of advanced mathematical content.
>
> ## Cons 3: ##
>
> Throughout this research, we explore the potential to combine the memory-type (Hopfield) neural network with mainstream method based on error propagation. The recent research on neuroscience reveals the surprising fact that the echo-location system of bats only consist of sixteen neurons as core module, which motivates us to exploit the potential of bio-inspired recurrent neural network. To implement this principle, we carefully begin with the modification of continuous HNN and specify a concrete task in robotics by embedding physical information with quaternion.
>
> The ideas and methods proposed in this paper do indeed originate from some of the content of HNN, but they are essentially  different from traditional HNN and are not included in any existing models. As mentioned earlier, we are not simply modifying HNN. Regarding the neural network model mentioned by the reviewer, our explanation is as follows:
>
> - Recurrent neural networks such as LSTM are actually completely different from Hopfield type associative memory. The former is merely capturing remote associations of sequence signals and has no direct connection with the operating principles revealed by neuroscience from hippocampus and amygdala in the human brain. More importantly, it does not equip with the capability to solve the benchmark problems mentioned in Cons 1.
>
> - Transformer and its derived control models (basically using the Transformer to predict the next action) do not provide any Lyapunov or exponential convergence guarantees. They rely on training data and optimization methods to achieve empirical robustness. QSHNN theoretically proves the global exponential stability under the constraint of set weights, continuously approximating the target from any initial conditions. This property is more suitable for the mentioned tasks. What's more, the consistency of geometric structure is still a significant problem for transformer models.
>
> Recently published research shows that the attention mechanism of Transformer is equivalent to a discrete modern Hopfield neural network. It achieves the same fixed-point iterative mechanism as the Hopfield energy function through weighted search of key-value pairs. From this perspective, each attention calculation in the Transformer is equivalent to performing a discrete attractor update on the memory state, and the update process relies on the real-valued vector space and the Euclidean dot product energy. In contrast, QSHNN establishes a continuous-time Hopfield dynamical system in the quaternion domain. It not only natively integrates the three-dimensional rotational geometric structure into the neural network but also strictly proves that the system has global exponential stability for a single target through Lyapunov theory. It ensures that any initial state can continuously and smoothly converge to the target quaternion over time. In addition, the training process of QSHNN adopts periodic projection, directly forcing the maintenance of the quaternion left-multiplication manifold structure each time the weights are updated, thereby obtaining an interpretable closed-loop control law. However, the attention weights of the Transformer are only indirectly trained through task-level losses, lacking explicit maintenance of dynamic stability and geometric structure. If a comparison is to be made in the same 6-degree-of-freedom alignment task in robotics, the Transformer can be regarded as a predictor of the joint velocity at the next moment, approaching the target through a certain number of attention layers or iterative updates, and then its convergence speed and computational steps can be compared with QSHNN. It is expected that although attention inherently possesses the characteristics of Hopfield networks, QSHNN, with its continuous dynamical structure for the target and periodic projection training, can definitely achieve higher accuracy, along with strict global stability guarantees.

---

> > ### Author Response · Authors · 2025-08-08
> >
> > Dear reviewer  wmYX,
> >
> > Thank you again for taking the time to review our paper. As the discussion phase will close shortly, please let us know if you have any further questions or comments on our rebuttal. We would be happy to provide any clarifications before the discussion ends.

---

### Official Review · Reviewer_7M5t · 2025-07-09

**Clarity:** 2
**Significance:** 3
**Originality:** 3
**Rating:** 5
**Confidence:** 2

**Summary:**

This paper presents for training a quaternion-valued Hopfield-structured neural network using supervised learning rules, in contrast to traditional unsupervised Hebbian learning. The goal is to ensure that the system asymptotically converges to specified target states. The network’s state evolves according to a quaternionic differential equation, and learning is guided so that the system stabilizes at externally defined equilibria. Conceptually, this means that for a given task, the network is trained to evolve and converge toward the corresponding target configuration over time.

**Questions:**

Major:
1. How  would you compare this studies with other trajectory optimization method? pros and cons? an alternative?
2. If direct encoding in early Hopfield-type networks lacks an error-driven optimization mechanism; thus preventing the network from refining its behavior in response to task-specific objectives; how does the supervised learning approach proposed in this work differ? It appears that the network still needs to be retrained whenever the target changes. In that case, how is this method more convenient or flexible than traditional approaches?
3. Lines 31–33: The sentence "The constants $\gamma$ and $\mu$ correspond to the parameters of electronic components capacitance and resistor for the circuit implementation of classic HNN." could be clarified further. As written, it is not clear what exactly you mean by electronic components capacitance and resistor.
4. The meaning of lines 155–158 is unclear and possibly incorrect. The statement says "X is noticeable as an anti-symmetric matrix plus a scalar times the identity matrix E, since the anti-symmetric matrix has only imaginary eigenvalues, which makes it easy to know X has two different conjugate complex eigenvalues, and they all have the same algebraic multiplicity, and the geometric multiplicity is 1". However, this is not generally true. For example, consider the anti-symmetric matrix: [0 -1 0 0; 1 0 0 0; 0 0 0 -1; 0 0 1 0]. This matrix has 2 complex eigenvalues, and each eigenvalue has a geometric multiplicity of 2.  Therefore, the claim in the original sentence does not hold in general and should be revised or further justified.
5. Please provide a more detailed explanation for Figure 9. It is not immediately clear what new information or insight Figure 9 is intended to convey compared to Figure 8.


Minor:
1. notation consistence. e.g. epsilon notation is not consistent.
2. Notation used without fair justification in section 3.

**Ethical Concerns:**

["NO or VERY MINOR ethics concerns only"]

**Final Justification:**

I agree with some of the other reviewers that it would be beneficial to include a discussion of the paper's limitations, either in the main text or in the appendix. Additionally, more detailed ablation studies and comparative experiments would help strengthen the paper. Nonetheless, I remain positive about the novelty of the work and believe it merits favorable consideration—especially with the additional experiments the authors proposed in the rebuttal.

**Limitations:**

yes

**Paper Formatting Concerns:**

No major formatting issue.

**Quality:**

2

**Strengths And Weaknesses:**

# Strength
**Quality**: This paper provides a thorough theoretical analysis of a quaternion-valued Hopfield-structured neural network and its supervised learning rule. The experimental results further demonstrate fast convergence and stability of the proposed learning scheme based on periodic projection.

**Clarity**: This paper provides a clear explanation on quaternion.

**Significance**:  The supervised learning rule introduced in this paper overcomes the lack of error-driven optimization in previous models, enabling the network to be trained to match externally defined targets.  The proposed model exhibits smooth dynamical behavior that is well-suited for applications such as robotics and trajectory generation.  By operating under a supervised learning paradigm with quaternion-valued trajectories, the network ensures that generated paths have bounded curvature, promoting trajectory smoothness.

**Originality**: This work is original in addressing the gap between traditional Hopfield-type networks and task-specific objectives. Moreover, by periodically projection onto the quaternionic structure, the training process remains more stable.

# Weaknesses
**Quality**: Although the periodic trajectory converges to the minimizer, the Frobenius orthogonal projection interrupts strict loss minimization, as acknowledged in the paper. Additional experiments would strengthen the validation of this projection strategy.

**Clarity**: some sentences are difficult to understand or ambiguous. (See the “Questions” section below for specific examples.)  Additionally, the paper lacks a clear explanation of the Hopfield Network framework, which is important for readers unfamiliar with the concept. I recommend adding a brief background section to improve accessibility.

**Significance**: The effectiveness of the method is not compared against other supervised trajectory generation/optimization approaches.

---

> ### Author Rebuttal · Authors · 2025-07-30
>
> Dear reviewer 7M5t:
>
> Thanks for taking your time reviewing our paper. Below are our responses:
>
> ## Weaknesses ##
>
> ### Quality: ###
> Regarding the application of periodic projection methods in neural network training over special algebraic domains. First of all, the concern raised by the reviewer is sensible. Weight mapping indeed disrupts strict gradient descent, but it does not disrupt the optimization process of the loss function and is currently the most suitable strategy. The reasons are as follows:
>
> - To integrate physical information into neural networks, the preservation and training of structures on special algebraic fields are crucial. However, as demonstrated in our paper, the calculus of quaternions, G$\mathbb{HR}$ calculus, is extremely complex and cumbersome. For other associative algebras that can be represented by matrices, there are not even rigorous calculus tools available at present. We do not wish to spend a considerable amount of time developing mathematical tools when expanding the model of this article to other types of problems. Moreover, absolutely strict formulas would also impose a heavy burden on calculations, significantly slowing down the response speed of the planning module when the target is updated.
>
> - When we integrate multiple neurons into a quaternion neuron, the weight parameters are redundant in terms of degrees of freedom, which will allow the optimized path to be explored in a high-dimensional space and can compensate for the interference of the periodic mapping maintaining the algebraic structure on the strict gradient. In the experiment, we set the training cut-off condition as $MSE<10^{-8}$ and have stably completed the training on a large number of randomly generated targets consistently. This verifies the capability of the periodic projection method in maintaining algebraic structure and optimizing effect. It can also be seen from the training trajectories of the illustration in the Figure (10) that this kind of interference does not seriously disrupt the path of strict gradient descent.
>
> Taken together, periodic Frobenius projection strikes the right balance between enforcing strict quaternion‐structure preservation and achieving efficient, error‐driven optimization in hypercomplex networks.
>
> ### Clarity: ###
> Throughout this research, we explore the potential to combine the memory-type (Hopfield) neural network with mainstream method based on error propagation. The recent research on neurosicence release the surprising fact that the echo-location system of bats only consist of sixteen neurons as core module, which motivate us to exploit the potential of bio-inspired recurrent neural network. To implement this principle, we carefully begin with the modification of continuous HNN and specify a concrete tasks in robotics by embedding physical information with quaternion. Thus our paper could not be treated in the classic Hopfield framework. For a quick understanding of the basic ideas of HNN, you could check reference [9]. We would consider adding more contents within the background section making the paper easier to follow.
>
> ### Significance: ###
> It is responded in Question 1.
>
> ## Question 1: ##
> Based on the theoretical foundation of the proposed model, we plan to develop a complete application for robotic manipulator planning in the follow-up publication. The following are the benchmark problems we consider, and baselines we choose to make comparisons.
>
> - **Robosuite PandaReach** (cited from reference: robosuite: A Modular Simulation Framework and Benchmark for Robot Learning, Y. Zhu (2020)) Robosuite's Franka Panda Reach mission only requires end-to-target position and pose alignment, which is closer to the real industrial scene. The environment has 6 degrees of freedom + claws, allowing us to focus on evaluating attitude control rather than grabbing strategies.
>
> - **OpenAI Gym FetchReach** (cited from reference: Multi-Goal Reinforcement Learning: Challenging Robotics Environments and Request for Research, M. Plappert(2018)) FetchReach is a 7-degree-of-freedom Fetch robotic arm target-alignment task. The target position is randomly generated in the environment (expandable to quaternions with attitude constraints), and the observation includes the relative position and direction of the end and the target, and the action directly gives the joint speed. In this environment, we can evaluate the response time, final error, and step time overhead of the QSHNN drive joint angle evolution to a specified quaternion pose.
>
> - **Random Workspace Target Alignment**, An Industrial Simulation (cited from reference: Convex and analytically-invertible dynamics with contacts and constraints: Theory and implementation in MuJoCo, E. Todorov (2014)) Build a UR5 or KUKA LBR model directly in MuJoCo, uniformly sample the attitude targets (including position and quaternion direction) in their workspace and test them on large number of random initial/target pairs.
>
> Through these three levels of benchmarking, we can cover the widely recognized standard environment in the academic community and verify the versatility and scalability of direct sampling targets in industrial simulation, thereby fully demonstrating the combined advantages of QSHNN in terms of response speed, control accuracy, and online computing cost.
>
> Respectively, the baselines are the performance of current industrial algorithms:
>
> - Damped Least-Squares Inverse Kinematics (**IK**): A closed-loop control law that solves joint increments directly based on Jacobian matrices and can be run online in milliseconds. It ensures local asymptotic convergence, but it is strongly dependent on the initial point, and often only achieves a low accuracy of rad.
>
> - Covariant Hamiltonian Optimization for Motion Planning (**CHOMP**): Optimized trajectory planning to generate smooth paths, but requires offline parameter adjustment and is time-consuming online
>
> - RRT-Connect + B-Spline Smoothing: First, use a fast random tree (**RRT**-Connect) to quickly generate a feasible path, and then use **B-Spline** or quadratic programming to smooth the path. This not only retains the efficient connectivity of the sampling algorithm, but also obtains a certain degree of smoothing effect, but the continuity and convergence lack strict guarantee.
>
> - Stochastic Trajectory Optimization for Motion Planning (**STOMP**): An iterative trajectory optimizer that samples noisy perturbations of an initial guess and weights them by smoothness and collision‑avoidance costs; while it generates low‑curvature paths, it depends heavily on the initial trajectory and requires tens to hundreds of milliseconds per planning episode, lacking global convergence guarantees.
>
> QSHNN has the potential to outperform these traditional strategies in terms of their weaknesses mentioned. In addition to the guarantees on dynamical properties, the powerful representation capability of QSHNN allows small-scale neural network to complete tasks whereby significantly improving the online planning reaction speed and reduce response time with the adjustment of target.
>
> ## Question 2: ##
> The traditional Hopfield neural network does not possess the planning capability mentioned in this paper. QSHNN is fundamentally different in the following aspects: HNN generates relatively vague multi-associative memory and lacks accurate and high-precision dynamic characteristics; QSHNN integrates the representation of quaternions on the rotation group SO(3) of rigid bodies, which can perfectly match the motion path planning task of robotic arms, a capability that HNN does not possess. In fact, in the tasks mentioned, past research does not developed a mature method mainly based on neural network. As for the issue of target updates, the small size and high representational ability of QSHNN can fully support real-time re-planning. Simultaneous multi-target planning itself is not the focus of such tasks. The benchmark problems and baselines we select in the subsequent research also meet these characteristics.
>
> ## Question 3: ##
> Hopfield-structured neural networks have a physical implementation by electronic circuits. Capacitance and resistor are common electronic components. There is a type of hardware module that can integrate this kind of neural networks to achieve a faster response speed than computer computation.
>
> ## Question 4: ##
> We appreciate the reminder of the reviewer. When anti-symmetric matrix is mentioned, we just talk about the specific matrix of the proposed formula rather than the general properties of linear algebra. The eigenvectors and eigenvalues of the proposed matrix is manually calculated and checked, thus would not influence the major deduction. We will correct this typo.
>
> ## Question 5 ##
> When we set the target quaternion in the form of $q_d=s_d+\boldsymbol{i}x_d+\boldsymbol{j}y_d+\boldsymbol{k}z_d$ with the extra condition $s_d=x_d=y_d=z_d$, the trained weights will concentrate on the main diagonal of the matrix. This concentration on the main diagonal reflects that, for an “equal‐component” target quaternion, the network need only scale the identity action in each block to achieve convergence. Figure 9 therefore highlights QSHNN’s ability to adapt its block‐wise quaternionic structure to target‐specific symmetries, a behaviour not apparent in the fully heterogeneous case of Figure 8.

---

> ### Comment · Reviewer_7M5t · 2025-08-01
>
> All of my questions have been addressed.

---

> > ### Comment · Reviewer_7M5t · 2025-08-05
> >
> > I agree with some of the other reviewers that it would be beneficial to include a discussion of the paper's limitations, either in the main text or in the appendix. Additionally, more detailed ablation studies and comparison experiments would help strengthen the paper. Nonetheless, I remain positive about the novelty of the work and believe it merits favorable consideration, especially with these improvements in place.

---

> > > ### Author Response · Authors · 2025-08-05
> > >
> > > Thank you for your constructive feedback. We agree that including a discussion of the paper’s limitations would strengthen the overall presentation. In the revised version, we will explicitly acknowledge the key limitations in the main text and outline potential directions for building complete applications, including comparisons with baseline methods on benchmark problems, which will be discussed further in the appendix.
> > >
> > > Regarding traditional ablation experiments, we believe they offer only limited interpretive value in our setting. The proposed model is derived entirely from a unified mathematical structure, without separate modules performing distinct tasks. Nevertheless, the spirit of ablation is already reflected in our experimental section, where we compare the learned weight matrices with and without the use of the periodic projection mechanism. Without this mechanism, the sub-blocks of the weight matrix lose their structured form, disrupting consistency within the quaternion field (Figure 7). In contrast, with the projection applied, the quaternionic structure is clearly preserved and maintained throughout training (Figure 8). We will clarify this point more explicitly in the main text.
> > >
> > > We sincerely appreciate your positive assessment of the novelty and value of our work, and we remain committed to incorporating these suggestions in our future research.

---

> > > > ### Comment · Reviewer_7M5t · 2025-08-08
> > > >
> > > > I will raise my score

---

### Official Review · Reviewer_zuPu · 2025-07-10

**Clarity:** 2
**Significance:** 2
**Originality:** 3
**Rating:** 4
**Confidence:** 2

**Summary:**

This paper presents a Hopfield-structured neural network for learning quaternion-based dynamical systems. The motivation comes from the use of quaternions in representing rotations and postures in various applications like robotic arm control. Inspired by the power of Hopfield neural networks in modeling dynamical structures, the authors propose a way to extend it to the quaternion domain. A generalized $\mathbb{HR}$ gradient descent with periodic projection is proposed to learn the parameters of the model. Theoretical analysis and empirical results are provided to demonstrate the performance of the proposed method, with a particular focus on its asymptotic stability.

**Questions:**

What are the advantages of learning in the quaternion domain compared to learning in other domains?

**Ethical Concerns:**

["NO or VERY MINOR ethics concerns only"]

**Final Justification:**

The rebuttal addressed the questions I had raised; therefore, I increased my score. I like the idea of using Hopfield neural networks to tackle the robotic manipulation problem, and the theoretical analysis supports their claims well. However, I would still suggest comparing the proposed method with other approaches that could be applied to this task, either theoretically or experimentally.

**Limitations:**

Please see the Weakness and Questions section.

**Quality:**

3

**Strengths And Weaknesses:**

**Strength**

The motivation for designing a neural network operating in the quaternion domain makes a lot of sense and could be valuable for real-world applications such as robot manipulation.

**Weakness**

1, The model needs to be re-trained for every new target state, which could be expensive in tasks where the space of possible target states is extremely large or even continuous.

2, Although asymptotic stability analysis is provided for the trained matrix $W$, the discussion of the error between its corresponding equilibrium and the true target state is missing. I believe an error bound for the accuracy is needed to rigorously demonstrate the theoretical performance of the proposed algorithm.

3, The paper lacks a comparison with other methods that could be used for the same task (e.g., other quaternion-valued methods or learning in a different domain other than the quaternion domain).

---

> ### Author Rebuttal · Authors · 2025-07-27
>
> Dear reviewer zuPu,
>
> Thanks for taking your time reviewing our paper. Below are our responses.
>
> ## Weakness 1 ##
> We hereby make more clarifications to facilitate better understanding of the scenarios of the QSHNN model. We have established the generalization ability on the configure space for the initial states rather than the target states. Concretely, the training process based on a unique target try to solve all the path planning problems from any initial posture of a robot manipulation. The evolution trajectories of QSHNN dynamics provide a complete solution of this planning problem rather than just retrieve single recorded memory. The development of this capability comes from practical concerns in the area of robotics, where the inverse-kinetics solving and planning of multi-joint manipulator is an extremely difficult problem. We would not expect a super powerful approach that can solve the planning task from one infinite set to another infinite set in one period of training. The mature modern Hopfield neural network does have the ability to keep multiple memories (targets), but it is still not possible to overlap the entire state space.
>
> ## Weakness 2 ##
> The rapid convergence of exact error between the target state and the network state is already guaranteed by exponential (asymptotic) stability under the framework of Lyapunov theory. Considering the background of the readers and to enhance rigorousness, here we supplement a deduction and will add it to the appendix. Recall the notations for the trajectory and target are $q(t)\in\mathbb{C}^2([0,+\infty])^{n}, q_d\in\mathbb{R}^{n}$, and the square error is expressed by:
> \begin{equation}
> E(t)=\frac{1}{2\gamma}||q(t)-q_d||^2
> \end{equation}
> Take the time derivative of $E(t)$, by the chain rule:
> \begin{align}
> \frac{dE(t)}{dt}&=\frac{1}{\gamma}[q(t)-q_d)]\cdot\frac{d}{dt}[q(t)-q_d)]\\\\
> &=\frac{1}{\gamma}[q(t)-q_d]^T[-\gamma \mathbb{I}+\mu WJ_{\phi}(q)][q(t)-q_d]\\\\
> &\leq -2\lambda E(t)
> \end{align}
> where the first equal simply comes from the evolution equation of QSHNN and the inequal comes from the direct computation in Appendix B Equation B.3, the coefficient $\lambda$ is defined by:
> \begin{equation}
> \lambda:=\gamma-\frac{\mu}{2}\sum_{i=1}^n(|w_{ji}|+|w_{ij}|)>0
> \end{equation}
> The inequalility above is guaranteed by Theorem 3.2. Hence, The error evolution over time satisfies:
> \begin{equation}
> ||q(t)-q_d||\leq ||q(0)-q_d||e^{-\lambda t}
> \end{equation}
> Thus, the exact distance between the network state and the target will exponentially descent to infinitesimal (E<<1) and be negligible in practical operation.
>
> If the reviewer means the error between the equilibrium of the QSHNN and the target, which is the loss function of the training process, the model is already verified on a large enough set of targets, and the training ends consistently with the criterion that MSE is less than $10^{-8}$ (see the threshold in Section 4). To mathematically rather than empirically discuss how the loss function of a general ML model will converge is an optimization problem, and it is not related to the stability and beyond the core scope of this research.
>
> ## Weakness 3 ##
> First, we would emphasise that this research is not about how to make a neural network quaternion-valued, but  exploring a totally new learning paradigm. More specifically, we try to combine the memory characteristics of Hopfield neural network and the mainstream learning approach based on error propagation. There is very little research which can be found but of great significance. It is more aligned with how the biological brain works where it also equipped with the inference type (e.g. MLP) and the memory type (Hopfield). And our research is more about theoretical validation and the implementation of this huge principle rather that to construct a successful application at current stage. To embed quaternion directly to the neural network rather than the others (i.g. complex number, other associative algebras...) is the most effective and natural way for the robotic task we consider. The existing methods for such tasks of robotics are not usually based on neural networks, such as probabilistic roadmap (PRM) and stochastic trajectory optimization (STOMP), QSHNN will definitely outperform them in terms of sensitivity and computational cost. We have done performed simulations with QSHNN for a robot, and we may consider developing a mature application and make a systematic evaluation in the future. It is separated from this paper or it will deviate from the theoretical deduction.
>
> ## Questions ##
> We consider to utilizing the evolution trajectory of QSHNN to solve the path planning and control problem for multi-joint manipulator in robotics, where a quaternion can directly represent the posture of a full freedom joint, or one has to conduct a series of projection and stretching to the $\mathbb{R}$ domain, which is unstable and inconsistent. The same reason also applied for other algebras. Quaternion in the proposed model is necessary for the given task, which embeds physical information to the neural network, serving as a foundation for further discussion of rotational invariance and interpretability. To develop the application in other scenarios, we could consider different algebraic structure, and it is noted that the techniques (periodic project, stability analysis) in our research is still valid as a framework.

---

> > ### Comment · Reviewer_zuPu · 2025-08-05
> >
> > Thanks for the detailed response - it addresses my questions. Therefore, I'm willing to increase my score.

---

> > > ### Author Response · Authors · 2025-08-05
> > >
> > > We are pleased to hear that our response addressed your concerns, and we sincerely appreciate your time and thoughtful feedback. Thanks again.

---

### Note · Authors · 2025-08-16

Dear reviewers, ACs and SACs,

We sincerely thank all reviewers and the area chair for their time, constructive feedback, and engagement in the discussion.

We are pleased that our responses have addressed most concerns, and we will incorporate the suggested improvements in the final version. In particular, we will add a dedicated limitations paragraph to describe the scope and boundaries of our approach, expand the discussion on future experimental designs including benchmark selection and baseline comparisons, and emphasize the distinctions between QSHNN and existing neural networks to help readers better understand its potential and application scenarios.

---

### Decision · Program_Chairs · 2025-09-17

**Decision:**

Accept (poster)

**Comment:**

This work presents a Hopfield-structured neural network that can be used to learn quaternion-based dynamical systems. This is motivated from the use of quaternions in the representation of postures in applications like robotic arm control. Inspired by how Hopfield networks have been successfully used in modeling dynamical structures, the authors propose a way to extend this to quaternion domains. The work also proposes gradient based methods for learning the parameters of the model. The paper is complemented with various theoretical analyses and empirical results. The reviewers agreed this work meets the bar for presentation at Neurips.